# Payload distribution and capacity of mRNA lipid nanoparticles

Sixuan Li [1,7], Yizong Hu [2,3,4,7] ✉, Andrew Li[3], Jinghan Lin[2,3], Kuangwen Hsieh [1], Zachary Schneiderman[2,5], Pengfei Zhang [3], Yining Zhu[2,3,4], Chenhu Qiu[2,6], Efrosini Kokkoli [2,5], Tza-Huei Wang [1,2,3] ✉ & Hai-Quan Mao [2,3,4,6] ✉

Lipid nanoparticles (LNPs) are effective vehicles to deliver mRNA vaccines and therapeutics. It has been challenging to assess mRNA packaging characteristics in LNPs, including payload distribution and capacity, which are critical to understanding structure-property-function relationships for further carrier development. Here, we report a method based on the multi-laser cylindrical illumination confocal spectroscopy (CICS) technique to examine mRNA and lipid contents in LNP formulations at the single-nanoparticle level. By differentiating unencapsulated mRNAs, empty LNPs and mRNA-loaded LNPs via coincidence analysis of fluorescent tags on different LNP components, and quantitatively resolving single-mRNA fluorescence, we reveal that a commonly referenced benchmark formulation using DLin-MC3 as the ionizable lipid contains mostly 2 mRNAs per loaded LNP with a presence of 40%–80% empty LNPs depending on the assembly conditions. Systematic analysis of different formulations with control variables reveals a kinetically controlled assembly mechanism that governs the payload distribution and capacity in LNPs. These results form the foundation for a holistic understanding of the molecular assembly of mRNA LNPs.

Lipid nanoparticles (LNPs) formulated from a mixture of an ionizable lipid, a helper lipid, cholesterol, a PEG lipid, and therapeutic nucleic acids have been shown to be potent and safe prophylactic vaccines and therapeutic delivery vehicles. For example, two mRNA vaccines against COVID-19 have received full FDA approval, and positive therapeutic outcomes were reported in a phase 1 clinical trial for transthyretin amyloidosis in which CRISPR-Cas9 mRNA and a single guide RNA were co-delivered to the liver[1]. Along with successful applications, there have been efforts in investigating the packaging characteristics of LNPs. Through experiments with siRNA-, mRNA-, and plasmid DNA (pDNA)-loaded LNPs, some features of these vehicles have been

reported previously, including the assembled structures[2–4], interior location of cargos[4,5], lipid compositions[6,7], and dynamic behaviors during the purification process[4,8]. They provide better understandings of the structure-property-function relationship that may direct further optimization of LNP designs.

A typical formulation process for mRNA LNPs starts with rapid mixing of an aqueous solution of mRNA and an alcohol solution of lipids at a pH, e.g., 4.0, that is substantially lower than the pKa of the ionizable lipid[9,10], which is typically ~6.5. Cryogenic transmission electron microscopy (Cryo-TEM) showed that different LNP species, vesicular or solid, are formed under this condition[8]. During dialysis

[1]Department of Mechanical Engineering, Johns Hopkins University, Baltimore, MD, USA. [2]Institute for NanoBioTechnology, Johns Hopkins University, Baltimore, MD, USA. [3]Department of Biomedical Engineering, Johns Hopkins University School of Medicine, Baltimore, MD, USA. [4]Translational Tissue Engineering Center, Johns Hopkins University School of Medicine, Baltimore, MD, USA. [5]Department of Chemical and Biomolecular Engineering, Johns Hopkins University, Baltimore, MD, USA. [6]Department of Materials Science and Engineering, Johns Hopkins University, Baltimore, MD, USA. [7]These authors contributed equally: Sixuan Li, Yizong Hu. ✉e-mail: yhu38@jhmi.edu; thwang@jhu.edu; hmao@jhu.edu

against a buffer at the physiological pH of 7.4, the ionizable lipids lose most of their positive charges (i.e., deprotonation) and form a hydrophobic, amorphous core, rendering an electron-dense appearance to all LNPs under cryo-TEM[4,6,8] (reproduced in our experiments as shown in Fig. 1g). The payload distribution and capacity of mRNA LNPs are important characteristics to assess, because they hint at molecular assembly mechanisms and influence pharmacodynamics, pharmacokinetics, and delivery efficiency[6,11,12]. However, Cryo-TEM, and other common nanoparticle characterization methods such as small-angle neutron scattering[6], NMR[13], and nanoparticle tracking analysis[14], could not effectively resolve these payload characteristics at the single-nanoparticle level, primarily due to difficulty in distinguishing empty LNPs from those with a payload[8,15] (Fig. 1g, i), and in quantifying mRNA molecules in mRNA-loaded LNPs. In contrast, fluorescence-based detection may be better suited to elucidating these properties[16].

We herein developed a multi-color fluorescence spectroscopic technique that integrated a single-molecule detection (SMD) platform, fluorescence coincidence analysis, and a quantitative fluorescence deconvolution algorithm for characterization of the payload

distribution and capacity of mRNA LNPs. The SMD platform, namely cylindrical illumination confocal spectroscopy (CICS), features a single-fluorophore sensitivity and ~100% mass detection efficiency[17,18] owing to uniform fluorescent excitations by its one-dimensional laser beam shaping. This flow-based technique allowed us to detect the entire nanoparticle population passing the detector. By fluorescently labeling different species in LNP formulations, and subsequently analyzing the coincidence of the single-particle fluorescence signals, we were able to differentiate all species in an mRNA LNP formulation. More importantly, we quantified the mRNA payload distribution and capacity at single-particle resolution through a deconvolution algorithm of the fluorescence signal distribution of mRNA-loaded LNPs against that of free mRNAs.

This technique was first applied to characterize a commonly referenced DLin-MC3-based mRNA LNP formulation in the literature (hereafter termed "the benchmark formulation"). Next, the effects of formulation parameters including the dose of PEG lipid, nitrogen-to-phosphate (N/P) ratio, mRNA concentration, and mRNA size on mRNA payload distribution and capacity, and relative helper lipid content were analyzed. Based on the data attained from these experiments, we

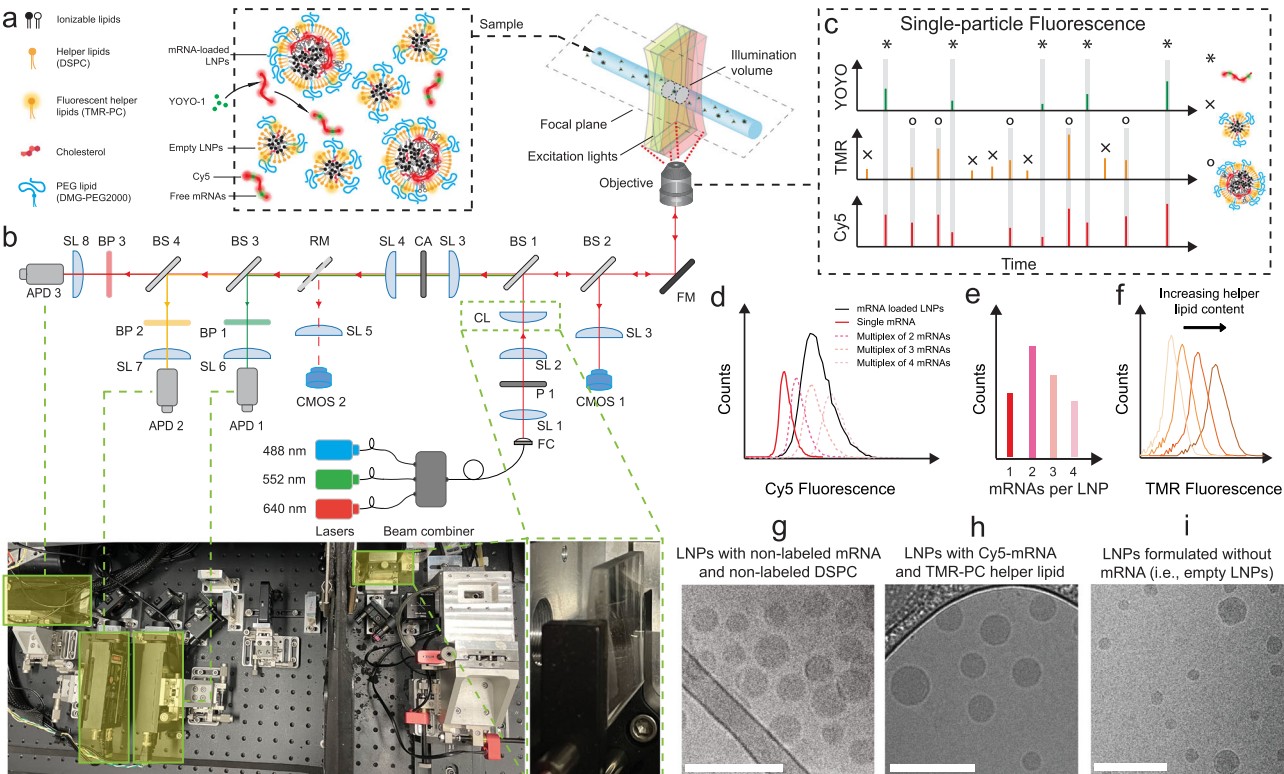

**Fig. 1 | Instrumentation of multi-color CICS platform, methodology for characterization of LNP formulations. a** Species of interest in LNP formulations include the mRNA-loaded LNPs, empty LNPs, and free mRNAs. Three fluorescent tags were used for the single-particle fluorescence detection and species classification: all mRNAs have Cy5 tags; 5% of the helper lipids carry a TMR tag; YOYO-1 was added into the LNP sample prior to CICS assessment to stain-free mRNAs. **b** Instrumental setup of three-color CICS. The lasers were first combined by a beam combiner to give a single output which allows the fluorescence coincidence detection, then expanded in one dimension by a cylindrical lens (CL) which gave an observation volume that covers the whole cross-section of the capillary. Such design allows CICS to obtain ~100% mass detection efficiency. The rectangular confocal aperture (CA) rejects the out-of-plane signal and confines the signal collection only from the center of the illumination volume, which renders highly uniform fluorescent signals. Each particle that passed through the detection volume generated a unique fluorescence signal that was recorded by single-photon counting avalanche photodiodes (APDs). **c** The single-particle fluorescence trace

was processed with a thresholding algorithm to identify all the burst events. Based on the fluorescent coincidence across the three colors, the fluorescence was classified as: mRNA-loaded LNPs (circles, TMR-Cy5 coincident), empty LNPs (crosses, TMR only), and free mRNAs (asterisks, Cy5-YOYO-1 coincident). **d** The Cy5 intensity profile of single free mRNA molecules, and theoretical Cy5 intensity profiles of multiplexed mRNAs expected in LNPs, compared with the histogram obtained from an LNP sample containing a distribution of the mRNA payload shown in **e**. **f** TMR intensity profiles of LNP formulations correlate with their relative helper lipid content. **g**–**i** Cryogenic transmission electron microscopy (cryo-TEM) images of mRNA LNPs of the benchmark formulation at pH 7.4, made **g** with non-labeled mRNA and non-labeled DSPC, **h** with Cy5-mRNA and 0.5% (mol% to total lipid content) TMR-PC, and **i** in absence of mRNA to form only empty LNPs. All scale bars = 200 nm. The images shown are representative images from two independent sample preparations and 50 TEM fields examined for each preparation, for which the findings were consistent.

**Table 1 | Differentiation strategy to distinguish different species in LNP formulations based on fluorescence coincidence analysis**

|        | mRNA-loaded LNPs | Empty LNPs | Free mRNA |
|--------|------------------|------------|-----------|
| YOYO-1 | –                | –          | +         |
| TMR    | +                | +          | –         |
| Cy5    | +                | –          | +         |

proposed a detailed kinetic assembly mechanism on how mRNA molecules are distributed into LNPs during the preparation process.

## Results

### Methodology, multi-laser CICS instrumentation, and deconvolution analysis

To track the mRNA in LNP samples (Fig. 1a), we used a commercially available Cy5-mRNA as the cargo that was 1929 nucleotides in length. As it was synthesized by substituting 25% of uridine to Cy5-uridine during RNA polymerization, the molecules have a statistical distribution of Cy5 copies per mRNA, reflected as a base Cy5 signal profile for single mRNAs (Fig. 1d). LNPs loaded with multiple mRNAs generate higher levels of Cy5 signal, representing ensembles of different numbers of mRNA molecules, reflected as right-shifted histograms. A fluorescently labeled helper lipid, TMR-PC, was added at a molar ratio of 0.5% to tag all LNPs. Statistically, LNPs with a higher content of the helper lipid (DSPC) are expected to carry more TMR-PC thus a higher TMR signal (Fig. 1f). We verified that the presence of fluorescent tags (on Cy5-mRNA and TMR-PC) did not perceptibly alter the size, the near-neutral nature of surface charge, the encapsulation efficiency (Supplementary Table 1) or the morphology (Fig. 1g, h) of the mRNA LNPs. A nucleic acid-intercalating, lipid-impermeable dye YOYO-1 was added prior to CICS assessments to specifically stain unencapsulated mRNAs (Supplementary Fig. 1).

The multi-color CICS platform was constructed as shown in Fig. 1b, (see Methods section for details). Concentration-optimized samples were introduced into a micron-sized capillary by a pressure-driven flow at a throughput of ~3000–5000 events/min that ensured one particle transits through the observation volume at a time. Three lasers with a wavelength matching the excitation spectra of fluorescent tags (488 nm, 552 nm, and 647 nm) were used for detection. The design of a cylindrical lens rendered a one-dimensional laser light sheet that covered the entire cross-section of the capillary, critical to a high fluorescence signal uniformity and mass detection efficiency[18,19]. When passing the detection window, each LNP or free mRNA generated a unique fluorescent burst signal, which was captured with single-fluorophore sensitivity by CICS[17,20]. The raw data were processed by a thresholding algorithm[21] to identify and quantify these fluorescent bursts. Different species of interest in an LNP formulation were determined by coincidence analysis of the fluorescence bursts (Fig. 1c, Table 1). Fluorescent spillovers across different channels were only occasionally observed in CICS (Supplementary Fig. 2). Compensations with single stain controls were carried out and proved to be effective for this CICS platform (Supplementary Discussion 1), even though we found that the quantification results to be reported throughout Figs. 2–7 were largely insensitive to compensation (Supplementary Table 5).

After identifying all mRNA-loaded LNPs, the mRNA payload in LNPs at the populational level can be estimated by comparing the mean Cy5 intensity of mRNA-loaded LNPs to that of the free mRNAs. However, the large variation in the fluorescence distribution prevents quantifying the payload for each LNP event. This variation is contributed by multiplicative factors[22,23] that are inherent in the measurement, including mRNA payload capacity, Cy5 copy per mRNA, Possionian nature of photon emission and detection, and fluctuation of laser power and flow rate. As the factors except for mRNA payload

capacity influence the measurement of LNPs and free mRNAs equally on CICS, it is then possible to quantify the mRNA payload capacity and its distribution by deconvolving the LNP Cy5 signal distribution against that of free mRNA (Fig. 1d). Detailed descriptions of the deconvolution analysis are in the Methods section. Briefly, the single-mRNA fluorescence distribution $D_{RNA,1}$ obtained by experiment was used to form the basis distributions $D_{RNA,n}|_{n=1,2,...,N}$, which was generated by multiplying the fluorescence of $D_{RNA,1}$ by $n$. $D_{RNA,n}$ represents the species of LNPs each containing exactly $n$ mRNA molecules. $D_{RNA,n}|_{n=1,2,...,N}$ was used to construct an estimated LNP distribution $D_{LNP}^*$ by assigning weights, $w_n$, to each basis distributions $D_{RNA,n}$.

$$D_{LNP}^* = \sum_{n=1}^{N} w_n \times D_{RNA,n} \qquad (1)$$

The experimentally obtained LNP distribution, $D_{LNP}$ was deconvoluted into a linear combination of these weighted base distributions. The weights added up to be the estimated total number of mRNA-loaded LNPs, $N^*$, which is the same as the experimental total number of mRNA-loaded LNPs $N$.

$$N = N^* = \sum_{n=1}^{N} w_n \qquad (2)$$

By tuning the weights to minimize the difference between $D_{LNP}^*$ and $D_{LNP}$, and an optimization factor given by $\chi^2$ (see Methods), the best fit $D_{LNP}^*$ was determined. The weights, $w_n$, in this best fit of $D_{LNP}^*$ describe the distribution of the number of mRNAs encapsulated in LNPs (Fig. 1e).

### Characterization of a benchmark mRNA LNP formulation

Using the aforementioned methodology, we characterized a benchmark formulation[24] prepared from a lipid mixture of DLin-MC3-DMA, 18:0 PC (DSPC), cholesterol, and DMG-PEG2000 at a molar ratio of 50:10:38.5:1.5. As the typical formulation process involves rapid mixing of lipids and mRNA solutions buffered at an acidic pH (in our study, 4.0) followed by dialysis against a buffer with pH 7.4, we sampled LNPs using three-color CICS at both pHs to reveal the detailed payload characteristics of LNPs before and after dialysis (Fig. 2a, b). Considering a highly over-charged state of fully protonated ionizable lipids at pH 4.0 (Supplementary Fig. 3), no YOYO-1 intercalation was observed (Fig. 2a). Plotting TMR vs. Cy5 signal intensities of all nanoparticle events allows clear identification of different populations in the LNP formulation. At pH 4.0 (Fig. 2c), three distinct populations were found: (1) TMR$^+$ Cy5$^+$ coincidences accounting for 34% of all events detected, which were presumably lipophilic mRNA complexes that contain a substantial amount of helper lipid. Note that the term "complexes" is used instead of "LNPs" to reflect the over-charged state; (2) TMR$^-$ Cy5$^+$ signals accounting for 25% of all events detected, suggesting they might be non-lipophilic, highly charged complexes of mRNA and ionizable lipids that could not accommodate as many helper lipids; (3) Cy5$^-$ TMR$^+$ signals accounting for 41% of all events detected, which were empty LNPs, i.e., LNPs without an mRNA payload.

Following dialysis (pH 7.4, Fig. 2d), the fraction of empty LNPs was more substantial, accounting for 77% of all LNPs. Cy5$^+$ TMR$^-$ events were unencapsulated mRNAs and accounted for only 4% of all events detected, which was expected for this formulation with a high encapsulation efficiency (Supplementary Fig. 4). The mRNA-loaded LNPs were identified by the coincidence of TMR and Cy5 signals. TMR-Cy5 labeling scheme (Fig. 2d) effectively distinguished mRNA-loaded LNPs against free mRNAs; Nonetheless, the three-color identification method with an additional YOYO-1 staining confirmation (Fig. 2e) was proved to be necessary to eliminate up to 10.4% of events found in the

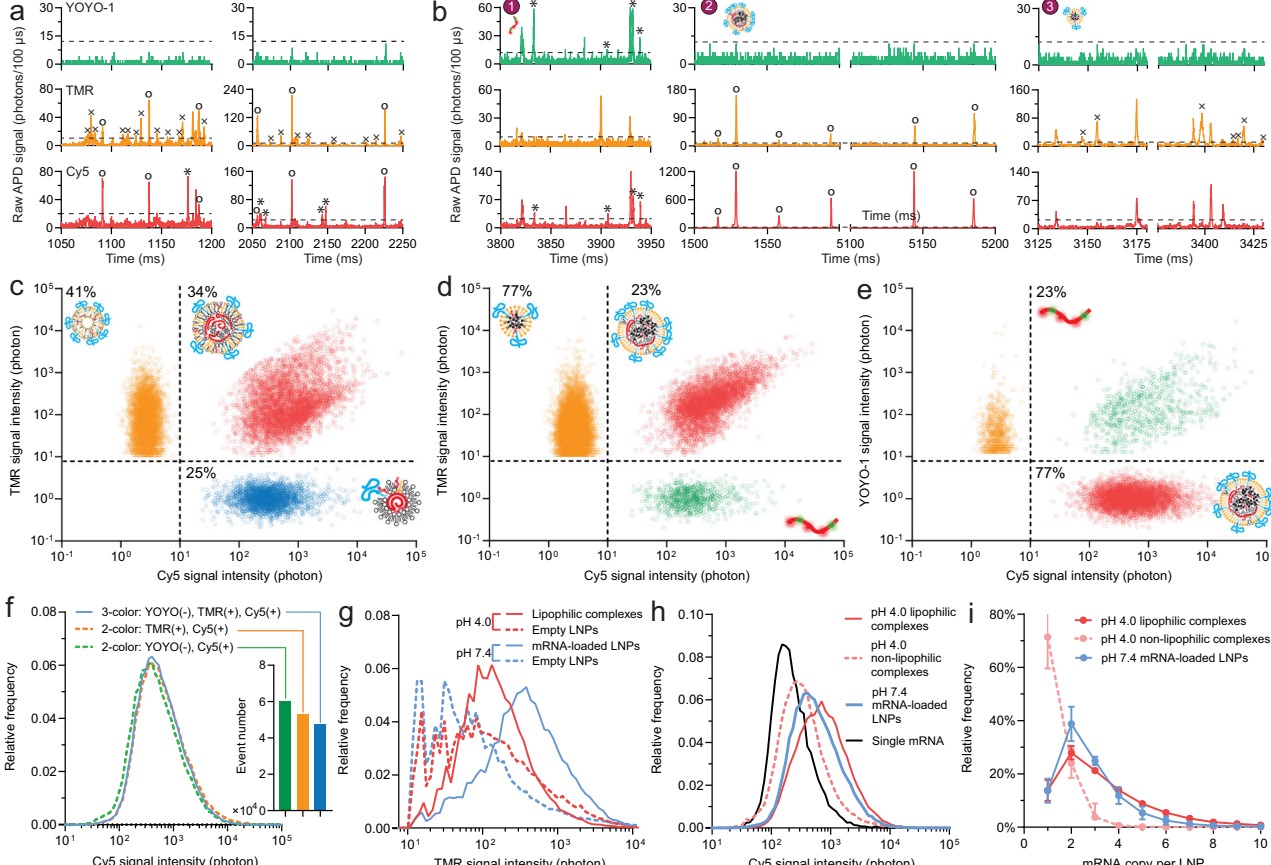

**Fig. 2 | mRNA payload behaviors of a benchmark mRNA LNP formulation (DLin-MC3-DMA: DSPC: cholesterol: DMG-PEG = 50:10:38.5:1.5). a** Example of 3-color raw signals at pH 4.0. Circles label events of lipophilic complexes; Asterisks label events of non-lipophilic complexes; crosses label events of empty LNPs. **b** Example of three-color raw signals upon dialysis to pH 7.4. Asterisks label events of free mRNAs in panel 1; Circles label events of mRNA-loaded LNPs in panel 2; Crosses label events of empty LNPs in panel 3. **a, b** The dashed lines show the threshold set for detection. **c** Classification of LNP species into empty LNPs (upper-left quadrant), lipophilic complexes (upper-right quadrant) and non-lipophilic complexes (lower-right quadrant) by plotting TMR signal intensity against Cy5 signal intensity at pH 4.0. 10% of 141,530 signals are shown in the figure. **d** Classification of LNP species into empty LNPs (upper-left quadrant), mRNA-loaded LNPs (upper-right quadrant), and free mRNAs (lower-right quadrant) detected at pH 7.4. For clarity, 10% of 195,090 signals are shown in the figure. The percentages labeled are relative to all

TMR events. Free mRNA events accounted for only 4% of all events. **e** Identification of mRNAs that were encapsulated in LNPs thus inaccessible to YOYO-1 and unencapsulated ones at pH 7.4 by plotting YOYO-1 signal intensity to Cy5 signal intensity. For clarity, 10% of 71,320 signals are shown in this figure. The upper-left quadrant population was presumably empty LNPs non-specifically tagged by YOYO-1. The percentages labeled are relative to all Cy5 events. **f** Application of three-color authentication for population classification reduced the frequency of false mRNA-loaded LNP signals from two-color authentications. **g** TMR signal intensity profiles of LNP species at pH 4.0 or 7.4. **h** Cy5 signal intensity profiles of single-mRNA molecules and LNP species at pH 4.0 or 7.4. **i** Calculated mRNA payload distributions of this benchmark mRNA LNP formulation using deconvolution algorithm ($n = 6$ independent formulation experiments). Data are presented as mean values ± SD.

two-color TMR/Cy5 labeling scheme that would have been falsely considered as mRNA-loaded LNPs (Fig. 2f). Therefore, we used the two-color identification method to achieve a higher degree of accuracy for calculating the payload distribution and capacity.

TMR fluorescence intensity profiles (as an indicator for relative helper lipid content, Fig. 2g) showed that lipophilic mRNA complexes at pH 4.0 or mRNA-loaded LNPs at pH 7.4 both contained a higher average helper lipid content than the empty LNPs. The comparison of the two conditions revealed a slight increase in helper lipid content from lipophilic mRNA complexes at pH 4.0 to mRNA-loaded LNPs at pH 7.4, corresponding to a decrease in helper lipid content in empty LNPs.

Using the deconvolution algorithm to analyze the fluorescence histograms of different LNP species (Fig. 2h), we depicted the mRNA payload distribution of the benchmark formulation (Fig. 2i). At pH 7.4, the number-average mRNA payload was 2.80 ± 0.41 among the mRNA-loaded LNPs, with around three-quarters of them carrying 1–3 mRNAs per LNP. Based on all the data collected with the three-color CICS experiment, a summary of the benchmark formulation is provided in Table 2.

## Effects of PEG lipid concentration on payload capacity of mRNA LNPs and composition drift during dialysis

Several reports demonstrated that the size of nucleic acid-loaded LNPs at pH 7.4 could be controlled by PEG lipid concentration[6,11,12]. The measured size of LNPs and a theoretical "size limit" were correlated when a critical molecular area was assigned to the PEG lipid at LNP surfaces[25]. When the mass content of other lipid components remains the same, a higher PEG% requires a higher surface-to-volume ratio, i.e., a smaller LNP size, to distribute the PEG lipids at a critical molecular surface density. In our experiments, by increasing the molar ratio of DMG-PEG2000 from 0.25% to 3%, the average LNP diameter at pH 7.4 decreased from 210 nm to 100 nm (Fig. 3a). The mRNA payload distributions of this LNP series analyzed by CICS clearly showed that the size difference directly correlates with the difference of payload capacity of mRNA LNPs (Fig. 3b, e). However, before dialysis at pH 4.0, the PEG concentration effect on LNP size was not observed (Fig. 3a), nor on mRNA payload distribution in either lipophilic or non-lipophilic mRNA complexes (Fig. 3c–e). These findings indicate that composition drifts occurred during dialysis due to deprotonation of ionizable lipids

**Table 2 | Composition features of the benchmark LNP formulation at an mRNA concentration of 20 μg/mL and an N/P ratio of 6**

| | Before dialysis at pH 4.0 (i.e., the initial LNPs) | After dialysis at pH 7.4 (i.e., the final LNP product) |
|---|---|---|
| Number-average payload (mRNA copy per particle) | Lipophilic complexes: $3.43 \pm 0.38$<br>Non-lipophilic complexes: $1.34 \pm 0.20$<br>All nanoparticles: $2.51 \pm 0.24$ | $2.80 \pm 0.41$ |
| Mode (most abundant) of mRNA payload | Lipophilic complexes: 2<br>Non-lipophilic complexes: 1 | 2 |
| Populations | $34\% \pm 8\%$ lipophilic complexes<br>$25\% \pm 4\%$ non-lipophilic complexes<br>$41\% \pm 10\%$ empty LNPs | $23\% \pm 8\%$ mRNA-loaded LNPs<br>$77\% \pm 8\%$ empty LNPs |
| Particle number concentration* | Lipophilic complexes:<br>$8.56 \times 10^{15} \pm 1.26 \times 10^{15}$ mL$^{-1}$<br>Non-lipophilic complexes:<br>$6.47 \times 10^{15} \pm 9.74 \times 10^{14}$ mL$^{-1}$<br>Empty LNPs:<br>$1.11 \times 10^{16} \pm 5.04 \times 10^{15}$ mL$^{-1}$ | mRNA-loaded LNPs:<br>$1.29 \times 10^{16} \pm 2.22 \times 10^{15}$ mL$^{-1}$<br>empty LNPs:<br>$4.88 \times 10^{16} \pm 2.49 \times 10^{16}$ mL$^{-1}$ |
| Encapsulation efficiency | N/A | $94.2\% \pm 3.6\%$ by RiboGreen**<br>$85.6\% \pm 5.1\%$ by CICS* |
| Average particle size*** | $106.3 \pm 13.0$ nm | $120.5 \pm 6.0$ nm |
| Zeta-potential*** | $+45.1 \pm 0.9$ mV | $-6.3 \pm 1.3$ mV |

Lipid composition: DLin-MC3-DMA:cholesterol:DSPC:DMG-PEG2000 = 50:38.5:10:1.5.

*The calculations for these parameters from CICS data are detailed in Supplementary Discussion 2–5;

**The assay is described in Methods;

***The particle size is reported as z-average diameter assessed by dynamic light scattering (DLS), that counted all empty or mRNA-loaded LNPs. The zeta-potential was assessed by phase analysis light scattering (PALS).

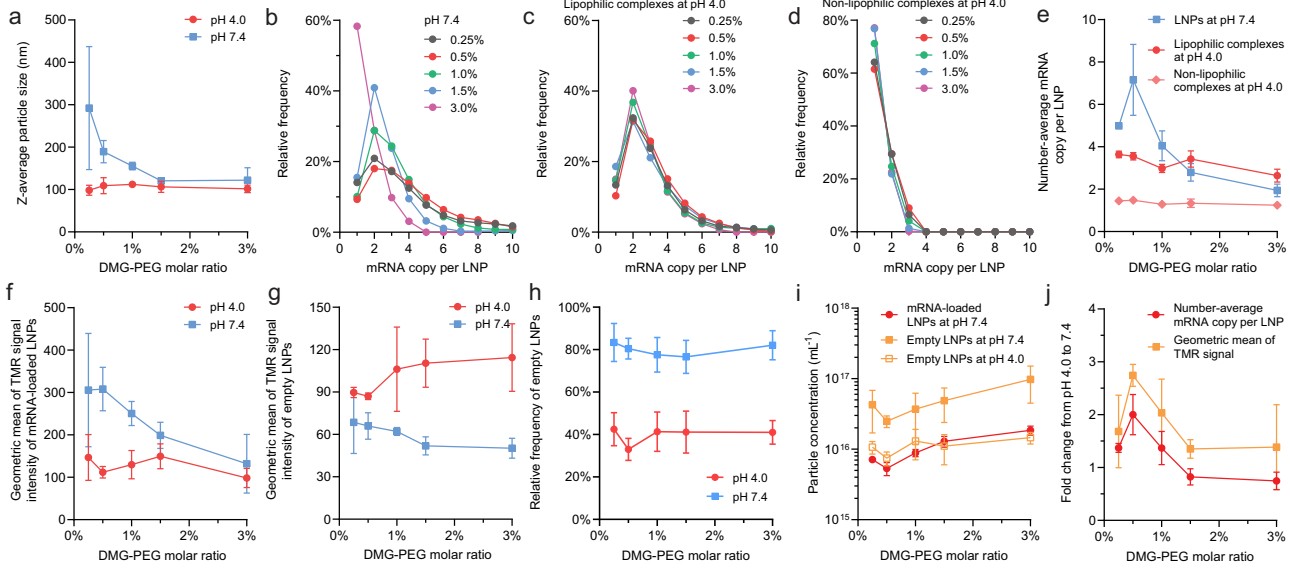

**Fig. 3 | Effects of molar ratio of DMG-PEG on the payload capacity and lipid content of mRNA LNPs (DLin-MC3-DMA: DSPC: cholesterol: DMG-PEG = 50:10:40-x:x). a** The z-average particle diameter of mRNA LNPs assessed by dynamic light scattering (DLS). **b–d** The mRNA payload distribution profiles of formulations at **b** pH 7.4; **c** pH 4.0 for lipophilic complexes; or **d** pH 4.0 for non-lipophilic complexes. **e** The number-average mRNA copy per LNP. **f, g** The geometric mean of TMR signals (indicator of relative helper lipid content) of **f** lipophilic complexes at pH 4.0 and mRNA-loaded LNPs at pH 7.4; or **g** empty LNPs at either pH 4.0 or 7.4. **h** The fraction of empty LNPs. **i** The absolute number concentrations of mRNA-loaded or empty LNPs at pH 7.4. **j** The average fold change of mRNA payload and helper lipid content from lipophilic complexes at pH 4.0 to mRNA-loaded LNPs at pH 7.4. The consistently higher fold change of helper lipid content indicated that merge of empty LNPs to lipophilic complexes occurred. **a, e, f–j**, data are represented as mean value ± SD, derived from $n = 3$ independent experiments (formulating LNPs from raw materials and then applying CICS analysis), except for 1.5% DMG-PEG where $n = 6$.

that transforms the LNPs from a state stabilized by surface PEG and an excess of residue positive surface charges at pH 4.0 to a state primarily stabilized by PEG at pH 7.4.

Our CICS data (Supplementary Fig. 5) comparing the states at pH 4.0 and 7.4 suggest that during dialysis, empty LNPs split (Fig. 4a-1, b-4) as indicated by a drop in their average TMR signal intensity (Fig. 3g) and an increase in their concentration (Fig. 3h, i). Many empty particles remained mRNA-free until being stabilized at pH 7.4 (Fig. 4a-2). Since the helper lipid DSPC was reported to

primarily reside on LNP surfaces[2,6], the driving force for splitting may be the transformation from a bilayer vesicle structure[8] at pH 4.0 to a single layer surrounding a hydrophobic core[4,8] (Fig. 1g) of neutralized lipids at pH 7.4. This conversion requires an extra surface area to distribute helper lipid and can be realized by splitting. At pH 4.0, lipophilic complexes carried more mRNAs per LNP than non-lipophilic complexes (Fig. 3c–e). When PEG% is high (e.g., ≥1.5%), some lipophilic complexes with a high initial payload split to give lower payloads during dialysis (Figs. 3b, c, 4a-3), whereas a

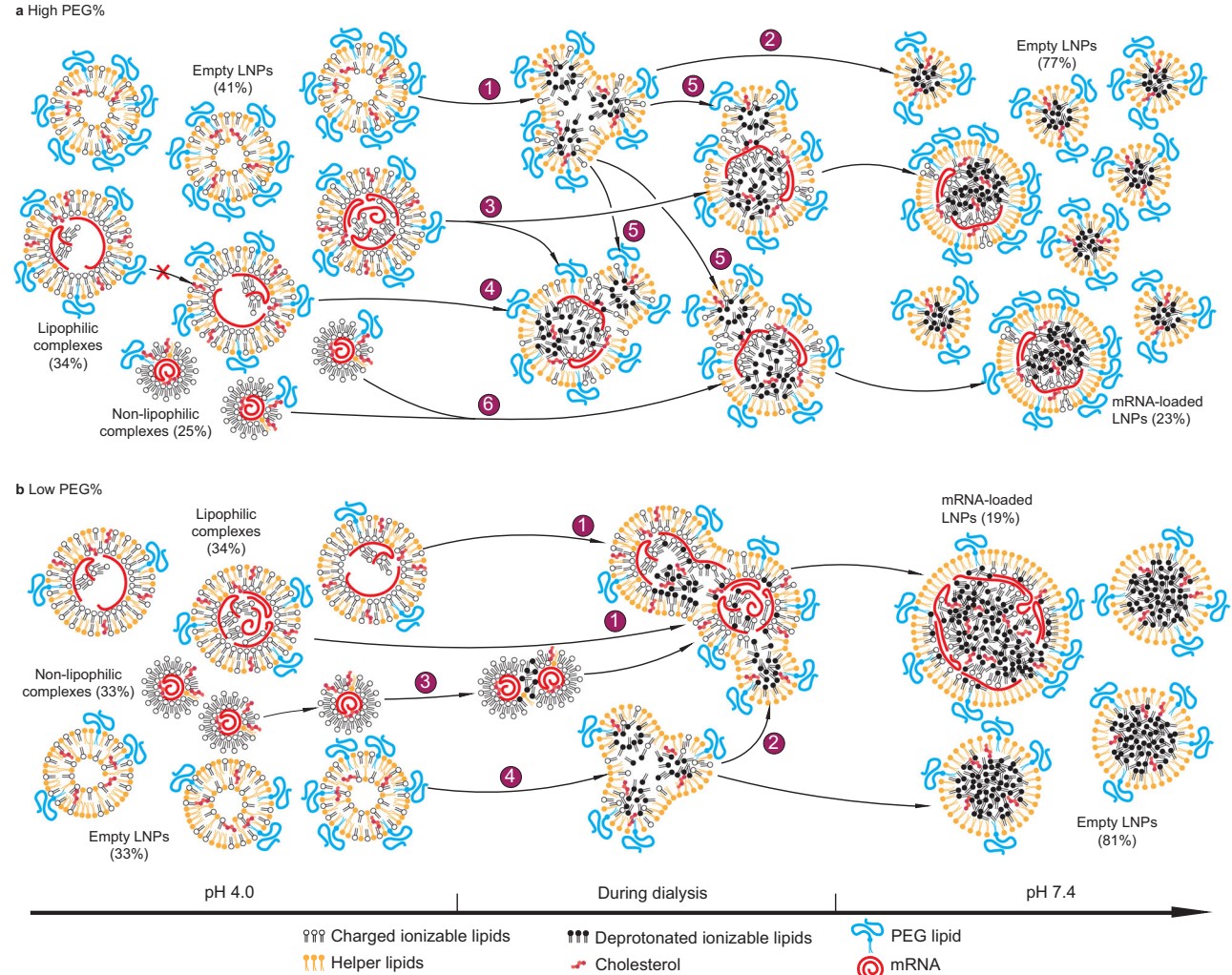

**Fig. 4 | Mechanisms of determination of payload capacity and distribution of mRNA LNPs by the PEG content. a**, **b** The hypothesized assembly processes and characteristics of LNP formulation with a high concentration of PEG mol% (**a**); or a low concentration of PEG mol% (**b**) and composition drift during dialysis from pH 4.0 (left) to pH 7.4 (right). The populational fractions labeled are real data from the formulation with PEG mol% = 1.5% (**a**) or 0.5% (**b**). **a** Each number label represents a populational behavior during dialysis: 1, splitting of empty LNPs; 2, stabilization of empty LNPs; 3, splitting of lipophilic complexes with an initially high mRNA payload; 4, remaining a same mRNA payload for lipophilic complexes with an initially low or intermediate payload; 5, merge of empty LNPs with mRNA complexes; 6, merge of non-lipophilic complexes. The cross mark represents the finding that the mRNA payload of lipophilic complexes does not increase during dialysis due to lack of merging under this condition. **b** The labels are: 1, merge between lipophilic complexes; 2, merge of empty LNPs with mRNA complexes; 3, merge of non-lipophilic complexes; 4, splitting of empty LNPs.

large fraction of them with a medium or low initial payload maintained the same payload (Figs. 3b, c, 4a-4). When PEG% is very high (e.g., ≥3.0%), splitting of lipophilic complexes becomes dominant thus resulting in a lower average payload (Fig. 3e) and a higher LNP concentration (Fig. 3i). These complexes received helper lipid content from merging with empty LNPs as indicated by an overall increase of TMR signal intensity after dialysis (Figs. 3f, j, 4a-5). This is presumably due to a lack of sufficient helper lipids and PEG lipids in the initial lipophilic complexes to stabilize these LNPs at pH 7.4. Because the payload distribution became relatively uniform at pH 7.4 (Figs. 2d, 3b), non-lipophilic mRNA complexes mostly carrying a single or two mRNAs at pH 4.0 must have merged during dialysis (Figs. 3b, d, 4a-6). At the same time, they originally did not contain any helper lipid (TMR⁻, Fig. 2c), thus they must have received it from empty LNPs during dialysis (Fig. 4a-5). These analyses are consistent with FRET and cryo-TEM observations in other reports[8], in which merging was considered the major event during dialysis.

A low PEG content (e.g., ≤1.0%) resulted in an increase in size limit at pH 7.4, enabling the lipophilic mRNA complexes to overcome the energy barrier to merge with each other (Fig. 4b-1), which significantly increased mRNA payload capacity (Fig. 3b, c, e). The LNPs received a significant amount of helper lipids from the empty LNPs during merging, as the fold increase of TMR signal was consistently found to be greater than that of mRNA payload after dialysis (Figs. 3j, 4b-2). Merging of non-lipophilic complexes (Fig. 4b-3) and splitting of empty LNPs (Fig. 4b-4) occurred in a similar manner as those with a higher PEG%.

### Effects of N/P ratio on payload capacity of mRNA LNPs and composition drift during dialysis

When N/P ratio (the molar ratio of amine groups on ionizable lipids to phosphate groups on mRNA) was tuned, the concentrations of all other lipid components were adjusted proportionally to that of ionizable lipid, while the mRNA concentration was kept consistent. This means that the PEG% to all lipids remained the same, yielding a consistent size of LNPs defined by the size limit correlated to the PEG surface density (Figs. 5a, 6). However, this same size permitted a higher mRNA payload per LNP as the N/P ratio decreased (Fig. 5b, e). The

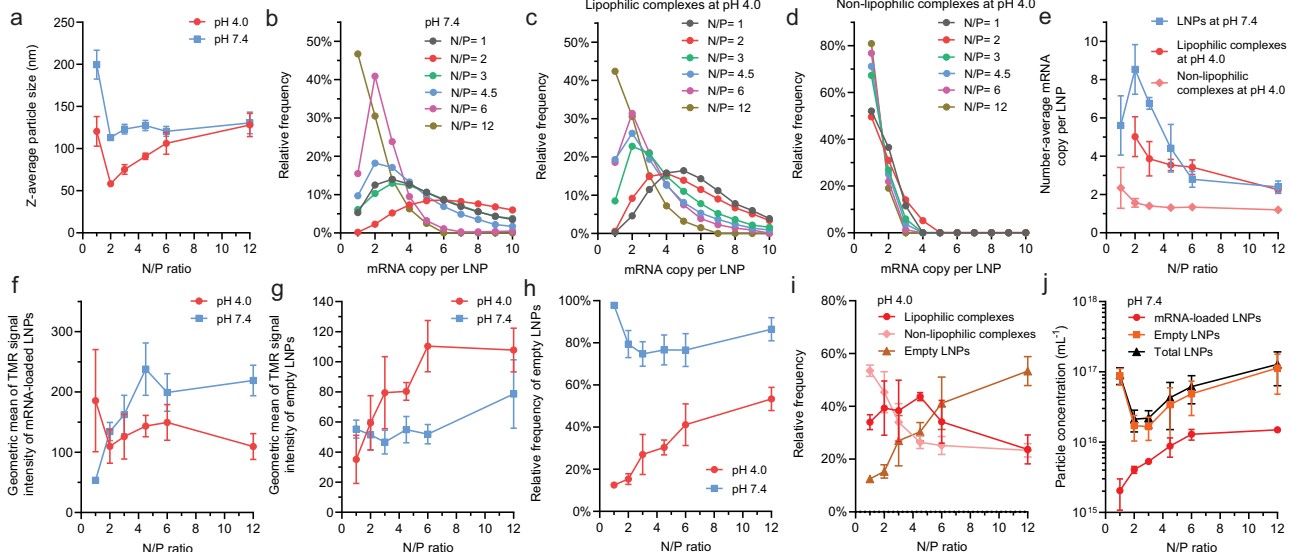

**Fig. 5 | Effects of N/P ratio on the payload capacity and lipid content of mRNA LNPs. a** The z-average particle size of mRNA LNPs assessed by dynamic light scattering (DLS). **b–d** The mRNA payload distribution profiles of formulations at pH 7.4 **b**; pH 4.0 for lipophilic complexes **c**; or pH 4.0 for non-lipophilic complexes **d**. **e** The number-average mRNA copy per LNP at either pH 4.0 or pH 7.4. **f, g** Geometric mean of TMR signals (indicator of relative helper lipid content) of lipophilic complexes at pH 4.0 and mRNA-loaded LNPs at pH 7.4 **f**, or empty LNPs at either pH 4.0 or 7.4 **g**. **h** The fraction of empty LNPs assessed at either pH 4.0 or 7.4. **i** The frequency of different LNP species at pH 4.0. **j** The absolute number concentration of LNPs at pH 7.4. **a, e, f–j** data are represented as mean value ± SD, derived from $n = 3$ independent experiments (formulating LNPs from raw materials and then applying CICS analysis), except for N/P ratio = 6 where $n = 6$.

dynamic behaviors of different species during dialysis were again found to be essential for payload determination (Fig. 5e, Supplementary Fig. 6).

As the N/P ratio decreases, the relative lipid-to-mRNA mass ratio decreases, reducing the relative ratio of lipid mass incorporated into mRNA complexes to that incorporated into empty LNPs at pH 4.0. This was reflected as a strong positive correlation between the fraction of empty LNPs and the N/P ratio (Fig. 5h, i), and a negative correlation for the fraction of mRNA complexes (Fig. 5i). Therefore, when N/P ratio is high (Fig. 6a), fusion of empty LNPs with mRNA-loaded LNPs is the kinetically favorable events (Fig. 6a-1) until the mRNA-loaded LNPs are fully stabilized at the size limit defined by PEG%. Fusion of mRNA-carrying complexes appears to be minor (Fig. 6a-2), and consequently the final stabilized LNPs contain relatively fewer mRNA payloads. When N/P ratio is low, mRNA-loaded complexes are surrounded by more mRNA complexes than empty LNPs (Fig. 5i), and fusion between non-lipophilic complexes and lipophilic complexes are kinetically favorable (Fig. 6b-1), whereas fusion of empty LNPs is less frequent (Fig. 6b-2). It is worth noting that mRNA complexes (lipophilic or non-lipophilic) with a lower N/P ratio also generally carried more copies of mRNA at pH 4.0 (Fig. 5c–e).

At pH 7.4, the mRNA-loaded LNPs at a higher N/P ratio contained a higher helper lipid content (Fig. 5f); while the empty LNPs shared a similar helper lipid content (Fig. 5g). A higher N/P ratio also generated a significantly higher concentration of LNPs (Fig. 5j).

### Effect of mRNA and lipid concentrations on payload capacity of mRNA LNPs

We next varied the mRNA concentration in the formulation from 5 to 100 µg/mL. The concentrations of all lipids were adjusted proportionally to maintain a constant relative lipid-to-mRNA mass ratio. Since the PEG% relative to all lipids remained constant, the same LNP size limit was observed at pH 7.4 (Fig. 7a). Therefore, the payload capacity and distribution profiles for LNP formulations in this series would remain the same; and this was verified by measured results as shown in Fig. 7a, b, and Supplementary Fig. 7. The helper lipid content of mRNA-loaded LNPs and the fraction of

empty LNPs appeared to be the lowest for the formulation with the lowest mRNA concentration of 5 µg/mL. Nonetheless, these metrics for all other formulations with 20–100 µg mRNA/mL were similar (Fig. 7c). At pH 4.0, these formulations yielded similar payload capacities of lipophilic and non-lipophilic complexes (Supplementary Fig. 8). This behavior was different from that of polyelectrolyte complexes of nucleic acids (e.g., pDNA/polyethyleneimine complexes), for which a higher overall nucleic acid concentration resulted in kinetic arrest of complexes with a higher pDNA payload per particle[26]. This difference may be explained by the high mobility of the cationic lipids as compared with polycations that ensures sufficient access to mRNA and charge neutralization, leading to effective formation of complexes with a lower degree of cross-complexation of multiple mRNA molecules.

### Payload distribution and capacity of mRNA LNPs with different mRNA sizes

We next tested LNP formulations with a smaller mRNA (996 nt, half of the first mRNA) and examined 3 assembly conditions (Supplementary Fig. 9): N/P = 3 and N/P = 6 as those discussed in Fig. 5; and 0.5% PEG as that discussed in Fig. 3. Formulation of LNPs with the same mRNA mass concentration means a doubled number concentration for this 996-nt mRNA. At pH 7.4, change of mRNA size did not significantly affect the size limit of LNPs as a result of the same PEG% (Fig. 7d); However, the average payload significantly increased (Fig. 7e, h). The two-fold reduction in mRNA size resulted in doubled payload in terms of the statistical mode (i.e., the most abundant) at pH 4.0 and 7.4 (Fig. 7f–h) for all LNP species. In the meantime, the helper lipid content of mRNA-loaded LNPs at pH 7.4 was similar between the two sets of LNPs with different mRNA sizes (Fig. 7i). These findings further support the conclusion that LNP assembly is most significantly influenced by the lipid concentrations or lipid-to-mRNA mass ratio, rather than mRNA concentrations. The copies of mRNA per LNP negatively correlated with the mRNA size, and it is approximated that the payload capacity of an LNP with a certain size limit can be reflected as a certain mass, instead of a certain number of mRNA.

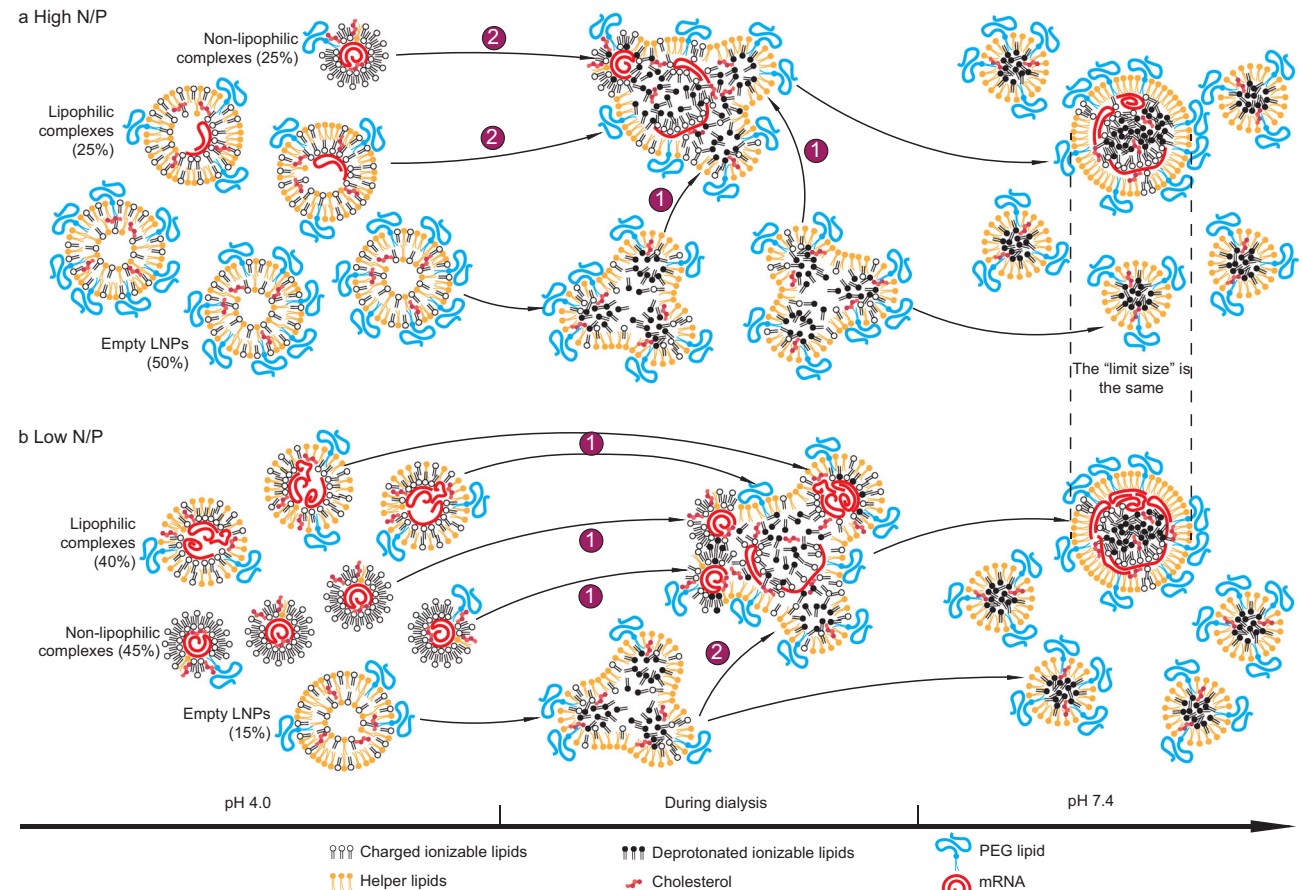

**Fig. 6 | Mechanisms of determination of payload capacity and distribution of mRNA LNPs by N/P ratio. a, b** The hypothesized assembly processes and characteristics of LNP formulation with a high N/P ratio **a**; or a low N/P ratio **b** and composition drift during dialysis from pH 4.0 (left) to pH 7.4 (right). At pH 4.0, the populational fractions labeled are real data for an N/P ratio of 12 (**a**) or 2 (**b**). Labels in both **a** and **b**: 1, a kinetically favorable (major) process; 2, a kinetically unfavorable (minor) process. At pH 7.4, the mRNA-loaded LNPs presumably hold the same size because the relative ratio of PEG lipid to all lipids is the same.

## Fraction of empty LNPs

Our analysis revealed that there was a significant fraction of empty LNPs in the final formulation at pH 7.4 for a wide range of conditions tested (Figs. 3h, 5h). We used an orthogonal method, super-resolution confocal microscopy imaging, to observe the benchmark LNP formulation embedded in solidified resin and verified this finding (Supplementary Fig. 10). A decrease in N/P ratio reduced the fraction of empty LNPs formed at pH 4.0 (Fig. 5h), highlighting the critical role of relative molar ratio of mRNA to lipids in determining the rate of lipid precipitation alone without mRNA, and the rates of mRNA-ionizable lipid complexation and concurrent lipid co-precipitation. However, the fraction of empty LNPs did not decrease with N/P ratio for these LNP formulations measured at pH 7.4. We attribute this to concurrent effects (Fig. 5j) of a decrease in the concentration of empty LNPs prepared at a lower N/P ratio due to less excessive lipids, and a decrease in the concentration of mRNA-loaded LNPs due to an increased mRNA payload (Fig. 6). When the mRNA size was reduced from 1929 nt to 996 nt, the fraction of empty LNPs decreased at both N/P = 3 and N/P = 6 (Fig. 7j), which agrees with the prediction from a previous report that larger nucleic acid cargo tends to result in higher fraction of empty LNPs[8]. With siRNA as cargo that was ~20 nt in size, a study found that there was no empty LNPs[15]. We hypothesized that smaller nucleic acid molecules with higher diffusivity facilitate better mixing with the ionizable lipids, leading to more uniform complexation and effective reduction in co-packaging of multiple mRNAs in a single LNP, thus reducing the fraction of empty LNPs.

This finding has important biological implications for LNP-mediated gene delivery as the majority of LNPs dosed do not carry an mRNA payload. It leads to unnecessary exposure to a high amount of lipid components for the body. To explore the role of empty LNPs in intravenous (i.v.) mRNA delivery, we prepared two LNP formulations at N/P ratios of 3 and 6, both with ~75% empty LNPs, and compared them with an LNP formulation from mixing LNPs prepared at N/P = 3 with additional empty LNPs to bring the total N/P ratio to 6 (contained 86% of empty LNPs; termed N/P = 3 + 3). Following i.v. injection of the three LNPs carrying luciferase mRNA as a reporter gene in Balb/c mice, the luciferase expression in the liver was reduced when the empty LNPs were added to the LNPs (Fig. 7k, l). The liver tropism of LNPs has been reported as a result of their interactions with apolipoprotein E (ApoE) in the blood[27,28], therefore the empty LNPs might have competed with mRNA-loaded LNPs for ApoE following injection and thus reduced delivery of mRNA to liver, though it did not alter the biodistribution to the liver (Supplementary Fig. 11a, c). When examining gene expression in the spleen, only the mRNA LNPs from N/P = 3 (both N/P = 3 and N/P = 3 + 3 groups), but not N/P = 6, generated appreciable gene expression (Fig. 7k, m) that might partially be attributed to their biodistribution profiles (Supplementary Fig. 11b, d). The mRNA-loaded LNPs in N/P = 6 held roughly half of mRNA payload comparing with those from N/P = 3, with slightly higher helper lipid content but the same size (Fig. 5). These results indicate that the fraction of empty LNPs and payload capacity may influence the transgene expression profile following i.v. administration, although further investigations are needed to provide mechanistic insights.

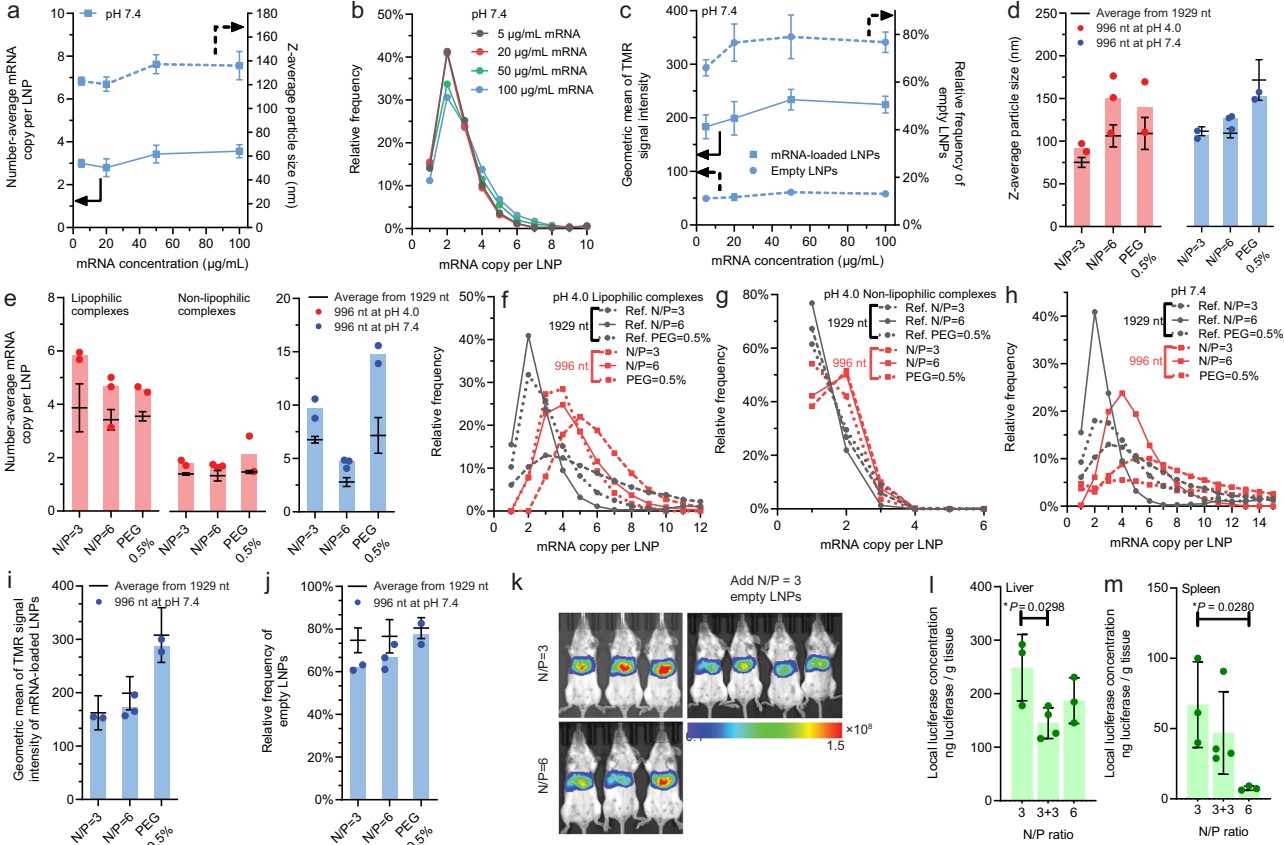

**Fig. 7 | Effects of lipid and mRNA concentrations and mRNA size on payload capacity and distribution, and effect of empty LNPs on mRNA delivery efficiency. a–c** The effect of mRNA (and lipids) concentration on (**a**) the z-average particle size and number-average mRNA payload ($n = 3$, except for 20 μg/mL where $n = 6$); **b** the payload distribution; and **c** the relative helper lipid content in mRNA-loaded or empty LNPs, as well as the fraction of empty LNPs at pH 7.4 ($n = 3$, except for 20 μg/mL where $n = 6$). **d–j** Effect of mRNA size (996 nt vs. 1929 nt) on **d** the z-average particle size, **e** the number-average mRNA payload, **f** the payload distribution at pH 4.0 for lipophilic complexes; **g** the payload distribution at pH 4.0 for non-lipophilic complexes; **h** the payload distribution at pH 7.4; **i** the relative helper lipid content of mRNA-loaded LNPs at pH 7.4; and **j** the fraction of empty LNPs at pH 7.4. **k–m** Effect of empty LNP content on mRNA delivery efficiency. The IVIS images of mice and harvested livers and spleens at 12 h post-i.v. injection of LNP formulations at an mRNA dose of 0.5 mg mRNA/kg. The harvested organs were

subsequently homogenized with the local luciferase concentration measured by ex vivo bioluminescence assay, and the results are shown in **i** for the liver and **m** for the spleen. **k–m**, N/P = 3 ($n = 3$) or 6 ($n = 3$) means the LNPs were directly formulated with an N/P ratio of 3 or 6, while N/P = 3 + 3 ($n = 4$) means the LNPs containing a lipid mass that equals to the mass correlating with an N/P ratio of 3 were added and mixed with the base. This N/P = 3 + 3 group contained ~2-fold of empty LNPs than the N/P = 3 group, while keeping the population of mRNA-loaded LNPs consistent. **k** The scale represents bioluminescence radiance with the unit of p/sec/cm²/sr. For statistically analysis in **l** and **m**, an unpaired, two-sided $t$ test was performed between the groups of N/P = 3 and N/P = 3 + 3, or between the groups of N/P = 3 and N/P = 6. **d–f**, **j** $n = 3$ independent formulation experiments for 1929 nt, except for N/P = 6 where $n = 6$. Data are presented as mean value ± SD throughout this figure.

## Discussion

We developed a single particle-analysis platform based on the CICS technique and reported for the first time the mRNA payload distribution and capacity, as well as the relative helper lipid content among different nanoparticle species in mRNA LNP formulations. This platform features high throughput and miniature sampling volume in a flow setup, which can be used for quality control purposes in the manufacturing mRNA LNPs (we showcased its application in assessing storage stability of LNP formulations in Supplementary Fig. 12) as well as fundamental mechanism studies. Using this method, we revealed that a benchmark mRNA LNP formulation contains mRNA-loaded LNPs mostly carrying 2 mRNAs in each particle with a number average of 2.8 mRNAs per LNP and contains ~80% empty LNPs at pH 7.4. We showed that the payload distribution and capacity are shaped by both the initial lipid phase separation and mRNA complexation at a low pH and compositional drifts during dialysis towards the physiological pH, in which the molar ratio of PEG lipids and lipid-to-mRNA mass ratio play a key role. The molar ratio of PEG lipids was found to dictate a size limit of the LNPs that positively correlated with the mRNA payloads, while

the lipid-to-mRNA mass ratio controls the fractions of the initial mRNA complexes vs. empty LNPs and kinetically influences LNP fusion. We also revealed that the payload distribution and capacity were insensitive to the concentrations of mRNA and lipids, while the payload capacity of an LNP formulation likely correlated with a certain mass of nucleic acids thus that each LNP would contain a higher copy number of cargos with a smaller cargo size.

Our work provides impetus for further studies. It demonstrated the feasibility to study complex nanoparticle systems through fluorescence detection modalities, and the methodology described might be well positioned to be adapted to high-performance imaging techniques, such as single-molecule-sensitivity total internal reflection fluorescence microscopy, for the assessment of payload features with structural information. Besides, it will be intriguing to determine the factors controlling the complexation processes in forming lipophilic and non-lipophilic LNPs upon lipid/mRNA mixing and controlling the initial payload distribution. It will be helpful to understand how lipid compositions (e.g., different structures of the ionizable lipids, species of helper lipids, and PEG lipids, and their relative ratios) influence the

mRNA payload distribution and capacity. In addition, it will be critical to understanding the effect of payload capacity and the fraction of empty LNPs on biodistribution, intracellular trafficking steps (e.g., cellular uptake, endosomal escape, cargo release), and mRNA expression kinetics. These directions will inspire further optimization of LNPs for the delivery of a wide range of nucleic acid therapeutics.

## Methods
### Preparations of mRNA LNPs
The lipids used in this study were the same as a benchmark LNP formulation[24]. The ionizable lipid DLin-MC3-DMA (MedKoo Biosciences, Cat# 555308), the helper lipid 18:0 PC (DSPC, Avanti Polar Lipids, Cat# 850365), cholesterol (Sigma-Aldrich, Cat# C8667), the PEG lipid DMG-PEG2000 (NOF America, Cat# GM020), and the fluorescent lipid TMR-PC (Avanti Polar Lipids, Cat# 810180) were dissolved in pure ethanol with a molar ratio of 50:10:38.5:1.5:0.5. Cy5-mRNA (TriLink Biotechnologies, Cat# L-7702 with a length of 1929 nucleotides or Cat #L-7701 with a length of 996 nucleotides) was dissolved in 25 mM sodium acetate buffer at pH 4.0 to be accounted for an mRNA concentration of 5–100 μg/mL in the final LNP product. The molar ratio of the amine groups on the ionizable lipid to the phosphate groups on mRNA (i.e., the N/P ratio) was kept from 1 to 12. When mRNA concentration was altered, the N/P ratio was kept at 6; when N/P ratio was altered, the mRNA concentration was kept at 20 μg/mL while the concentrations of DSPC, cholesterol, and DMG-PEG were altered proportionally to DLin-MC3-DMA. In the representative formulation (Fig. 2), the final mRNA concentration was 20 μg/mL with an N/P ratio of 6, correlating with 29 μg mRNA per μmol of total lipid components (including cholesterol). For formulating LNPs, a T-junction (IDEX Health and Science, Cat# P-890) was used. The lipid ethanol solution and the mRNA aqueous solution were injected into the T junction at a flow rate of 1 mL/min and 3 mL/min, respectively, controlled by two syringe pumps (New Era Pump Systems, Cat# NE-4000). The collected LNP suspension was dialyzed against 100-fold volume of 25 mM sodium acetate buffer at pH 4.0 (to remove ethanol) or phosphate-buffered saline (PBS) at pH 7.4 (to remove ethanol and raise the pH to physiological pH) for 12 h under 4 °C by tubings with a molecular weight cut-off (MWCO) of 3,500 (Pur-A-Lyzer dialysis kit, Sigma-Aldrich, Cat# PURD35050). The LNPs were characterized immediately following dialysis.

### Characterization of the size, zeta-potential and encapsulation efficiency of mRNA LNP formulations and optimization of YOYO-1 binding to free mRNA
Following dialysis, the size of the LNP formulations were assessed by dynamic light scattering (DLS, Malvern Zetasizer ZS90). The zeta-potential and the z-average diameter were reported for each formulation in this study. Quant-it™ RiboGreen assay (ThermoFisher Scientific, Cat# R11490) was used to characterize the encapsulation efficiency of the LNP formulations. Briefly, LNPs treated by 0.5% w/v Triton X-100 (Sigma-Aldrich, Cat# T8787) to distrupt LNP structure and release mRNA and untreated LNPs were diluted to a concentration below 1 μg mRNA/mL, and then reacted with equal volume of Ribo-Green assay solution at a 200-fold dilution. Standard curves were generated within 0.1 to 1.0 μg mRNA/mL using a series of free mRNA solutions with or without 0.5% w/v Triton X-100. The concentrations of free mRNA and total mRNA in the formulation were determined using bulk fluorescent reading (excitation: 480 nm, emission: 520 nm) of the sample against the corresponding standard curves. In CICS experiments, YOYO-1 iodide (ThermoFisher Scientific, Cat# Y3601) was used to stain unencapsulated mRNAs. To ensure highest detection sensitivity, the ionic strength of the PBS buffer at pH 7.4 and the molar ratio between YOYO-1 and mRNA were screened to yield a sensitivity over 95% (Supplementary Fig. 1), with 0.25-fold PBS and 1 nM YOYO-1 per 5 ng mRNA/mL being optimal, respectively. YOYO-1 was also used to

characterize the encapsulation efficiency of LNP formulations. Different from RiboGreen, in which a fixed 200-fold diluted working assay solution was used for staining, the YOYO-1-to-mRNA ratio was kept consistent as 1 nM per 5 ng for both the sample and standard curves. The encapsulation efficiencies determined was found to be similar as those determined by RiboGreen (Supplementary Fig. 4).

### Cryogenic transmission electron microscopy (cryo-TEM) of mRNA LNPs
The mRNA LNPs at pH 7.4 were first concentrated to a total lipid mass concentration of ~10 mg/mL using Vivaspin Protein Concentrator Spin Columns with a MWCO of 100,000 Da (Sartorius, Cat# VS0141). 5 μL of the concentrated LNP sample was deposited onto copper grids coated by a lacey carbon film (Electron Microscopy Services, Cat# LC200-CU) that had been treated with glow discharge. The sample was then vitrified by plunging into liquid ethane operated on a Vitrobot (Thermo Fisher Scientific). The Vitrobot parameters were as follows: blot time = 3 seconds, blot force = 0, offset = 0, wait time = 3 seconds, relax time = 0, and humidity = 100%. After vitrification, the grids were kept under liquid nitrogen and were transferred to a F200C Talos transmission electron microscope (Thermo Fisher Scientific) operated at an acceleration voltage of 200 kV. Images were captured using a Ceta camera equipped on the Talos instrument.

### Multi-color CICS instrumentaion
The 3-color CICS was an expansion of the previous single-color version[18]. A schematic of the optical setup is shown in Fig. 1a. The system contains three continuous wave lasers, with emission at 488 nm, 552 nm, and 640 nm (OBIS LS 488-100FP, LS 552-80FP, LS 640-75FP, Coherent). The three lasers go through a laser beam combiner (OBIS Galaxy, Coherent) and output a single beam after an achromatic fiber collimator ($f$ = 4.0 mm, Thorlabs). The beam is expanded by a Keplerian beam expander which consists of two achromatic doublets (SL1, $f_1$ = 19 mm and SL2, $f_2$ = 75 mm, Thorlabs) and a 50-μm pinhole (Lenox Laser). The beam is further expanded in one dimension by a cylinderical lens (CL, $f$ = 150 mm, Thorlabs), and a dichroic mirror (FM, Thorlabs) is used to direct the excitation light into a ×100 oil immersed objective (NA = 1.3, Olympus) which also collects the emitted fluorescent signal from the sample. The sample is transported by a pressure-driven flow in a fused silica microcapillary (Inner diameter = 10 μm, Molex). The capillary is cut to be 50 cm in length and a transparent observation window is made by burning the polyimide coating on the exterior of the capillary at the length of 45 cm from the sample inlet. The capillary is mounted onto a glass slide and then placed onto a custom-made sample stage, which is further mounted onto a moterized XYZ stage (9063-XYZ-PPP-M, Newport). Two dichroic mirrors, DM1 and DM2 (LM01-552-25 and BLP01-635R-25, Semrock) are used to separate the signals induced by the three lasers. Then, the signals pass through a rectangular confocal aperture (CA, 292 μm × 75 μm, National Aperture), which rejects the out-of-plane signal, and go through corresponding bandpass emission filters BP1, BP2, BP3 (FF02-520/28-25, FF03-575/25-25, and FF01-676/37-25, Semrock). The beams are then focused by doublets (SL5, SL6, SL7, $f$ = 30 mm, Thorlabs) onto the single-photon counting avalanche photodiodes (APD, SPCM-AQRH10, Excelitas). Two CMOS cameras (DCC3240C, Thorlabs) are used to accurately align the detection window to the microcapillary channel. A fraction of the light from the sample is directed by a pellicle beamsplitter (BS 2) and focused (SL 3) onto the first camera (CMOS 1) which guides the proper focus of the capillary. After the confocal aperture (CA), the second camera (CMOS 2) is used to acurately align the capillary position to be in the middle of the rectangular aperature, when the removable mirror (RM) is in place. During the experiment, a motorized flipper mount (8892-K-M, Newport) is used to switch off the mirror and direct the light to the APDs for data recording. A DAQ card (NI USB-6341, National Instruments) and a

custom LabVIEW (Version 2020, National Instruments) is used for data acquisition at a rate of 250 kHz, with a bin size of 0.1 ms. The data analysis is performed on a laptop with custom MATLAB codes (Version 2021a, MathWorks).

## Multi-color CICS experimental procedure

The LNP samples after dialysis in both sodium acetate buffer at pH 4.0 and PBS at pH 7.4 were further diluted in the corresponding buffer with 2% w/v PEG (20 kDa MW, Sigma-Aldrich, Cat# 81300). PEG was used as a dynamic coating additive to minimize adsorption in the capillary. After the encapsulation efficiency was determined, the free mRNA in the samples were stained with YOYO-1 iodide at a ratio of 1 nM YOYO-1 per 5 ng mRNA/mL. The mixture was incubated in PCR tubes in dark for at least 1 hour. The sample vial was placed in a pressure chamber and connected to the inlet end of the capillary. The sample was then injected into the capillary driven by a high-pressure argon gas (AR HP6K, Airgas) at 42 psi, which gave a flow rate of 1 mm/s. For each LNP formulation, at least 50,000 signals were collected over a 20-min period for data analysis. A sample of free mRNA stained by YOYO-1 at a ratio of 1 nM YOYO-1 per 5 ng mRNA/mL in both pH 4.0 and pH 7.4 were analyzed to collect the basis Cy5 signal histogram for individual single mRNAs. After each sample run, the capillary was cleaned by flushing 0.1 M NaOH, deionized water, and 0.1 M HCl three times, followed by the corresponding sample buffer. Each capillary cleaning run went through at least five capillary lengths at 800 psi.

## Single-nanoparticle analysis of mRNA LNPs using multi-color CICS

The single-nanoparticle data analysis of CICS consists of three parts: single-particle fluorescence burst quantification, three-color coincidence detection for particle classification, and deconvolution analysis for mRNA payload characterization. The first part, single-particle fluorescence burst quantification, has been described in detail in our previous works[18,20]: The raw single fluorescence data were processed by a thresholding algorithm to identify the single-nanoparticle burst events. The information of each burst event including the retention time (ms), the start and end time of the burst (ms), burst height (photons/ms), burst width (ms), and burst size (photons) were recorded. These identified bursts in each color went through a coincidence detection algorithm which matched the coincidence events in two colors. The algorithm selected the burst events with their retention time difference between the two colors smaller than half of their maximum burst widths. The algorithm went through all the two-color combinations including Cy5-TMR, Cy5-YOYO-1, and TMR-YOYO-1. The burst size of the identified coincident events was adjusted by the compensation matrix before output for the analysis next step. For the sample in pH 4.0 buffer, the Cy5-TMR coincidence events were classified as lipophilic mRNA complexes; Cy5 without TMR events were classified as the non-lipophilic complexes; TMR without Cy5 events were classified as the empty LNPs. For the sample in pH 7.4 buffer, the Cy5-TMR coincident events without YOYO-1 were classified as mRNA-loaded LNPs; the Cy5-YOYO-1 coincident without TMR were classified as free mRNAs; the TMR without any coincident events were classified as the empty LNPs; the TMR-YOYO-1 coincident events were classified as the non-specific binding YOYO-1. The fluorescence distributions of the classified species were plotted and analyzed by their geometric mean. The distribution of the LNPs was further used for the payload capacity analysis by deconvolution algorithm. Prism (Version 9) from GraphPad was used to plot most figures. LNP illustrations in Figs. 1, 4, and 6 were created by Adobe Illustrator.

## Deconvolution of CICS signals to correlate with mRNA payload in LNPs

The fluorescence signal deconvolution algorithm was first proposed by Mutch et al.[22,23] to process total internal reflection fluorescence

microscopic images as a way to count protein number. In our previous work[21], we applied this analytical tool to quantify the DNA content distributions in PEI/DNA and PEI-g-PEG/DNA nanoparticles, and the same method was adopted for the mRNA payload analysis in this work. First, the fluorescence distributions of the mRNA-loaded LNP species ($D_{LNP}$) and the free mRNA ($D_{RNA}$) were obtained by the CICS experiments and single-nanoparticle analysis, and normalized by their total number of events. The distributions of the particle fluorescence were best described by a lognormal rather than the Gaussian distribution explained by the multiplicative processes[22,23], and thus quantified with logrithmic binning. By multiplying $D_{RNA}$ by a scaling factor ($n$), a set of basis distributions $D_{RNA,n}|_{n=1,2,...,N}$ were generated, which essentially represents a set of monodispersed particles each containing exactly $n$ mRNA molecules. To maximize the computation accuracy, the upper limit ($N_{max}$) of the scaling factor was chosen to be six times the average number of mRNA per particle, which was estimated by the ratio of the geometric mean of fluorescence distribution of the mRNA-loaded LNPs to that of the free mRNAs.

$$\sum_{i=1}^{I_B} D_{RNA,n}(i) = 1|_{n=1,2,...,N} \tag{3}$$

$$N_{max} = 6 * \frac{\mu_{LNP}}{\mu_{mRNA}} \tag{4}$$

$D_{RNA,n}(i)$ represents the proportion of each distribution in $i$th bin, for all $n$. $I_B$ is the number of bins for each distribution.

$$D_{LNP}^* = \sum_{n=1}^{N} w_n \times D_{RNA,n} \tag{5}$$

$$0 \leq w_n \leq n_{LNP} \tag{6}$$

A fitted estimate mRNA LNP distribution, $D_{LNP}^*$ is constructed by assigning weights, $w_n$, to each basis distributions $D_{RNA,n}$, whereas $D_{LNP}$ is the mRNA LNP distribution obtained experimentally which is deconvoluted into a linear combination of the weighted basis distributions.

For each single bin, we have

$$y_i^* = D_{LNP}^*(i) = \sum_{n=1}^{N} w_n \times D_{RNA,n}(i) \tag{7}$$

where $y_i^*$ is the estimated number of the mRNA-loaded LNPs in the $i$th bin. And the estimated total number of mRNA-loaded LNPs, $N^*$, is given by

$$N^* = \sum_{n=1}^{N} w_n \tag{8}$$

The difference between the constructed mRNA LNP distribution, $D_{LNP}^*$, and the experimental $D_{LNP}$ is $\chi^2$.

$$\chi^2 = \left( \sum_{i=1}^{I_B} \frac{(y_i - y_i^*)^2}{y_i} \right) + \alpha \left( N_{LNP} - N^* \right)^2 \tag{9}$$

where $\alpha$ is a penalty factor imposed in the optimization. $\alpha$ was chosen to be 0.1 to ensure $N^*$ is <1% off from $N_{LNP}$. The goal of the deconvolution analysis is to minimize $\chi^2$ by finding the best weights assigned to the basis distributions, $w_n$ to describe the mRNA payload in the LNP sample. This optimization was performed by a simulated annealing algorithm in Matlab. All the source codes of the aforementioned data analysis can be provided upon request.

## Animal model

Mice were supplied free choice with pelleted feed containing low fiber (5%), protein (20%), and fat (5–10%), and water using automatic waterers. Mouse rooms were maintained at 30–70% relative humidity and a temperature of 18–26 °C (64–79 °F). The mice were housed in standard shoebox cages with filter tops, and provided with corncob as bedding. A 14 h/10 h light/dark cycle was maintained. Balb/cJ mice (female, 8 weeks old, The Jackson Laboratory) were used. The intravenous (i.v.) LNP delivery experiments were approved by the Johns Hopkins Animal Care and Use Committee (ACUC, protocol #MO20E63). The LNP formulations carrying luciferase mRNA (TriLink Biotechnologies, Cat# L-7202) were formulated as described above and dialyzed into PBS buffer, and subsequently injected through the tail lateral vein at a dose of 0.5 mg mRNA/kg. At 12-hour post-injection, 100 μL of D-luciferin solution (25 mg/mL in PBS, Gold Biotechnology, Cat# LUCK) was intraperitoneally injected into each mouse. The mice were imaged by an IVIS live-animal imaging system (Perkin Elmer) 5 min after injection. The liver and spleen were then harvested, weighted, and disgested by reporter lysis buffer (Promega, Cat# E4030) assisted by an ultrasonic processor (Qsonica, Cat# Q55A). The digested solution was subjected to a freeze-thaw cycle to fully release luciferase. The luciferase concentration within each organ sample was characterized by a standard luciferase assay (Promega, Cat# 1500).

## Reporting summary

Further information on research design is available in the Nature Research Reporting Summary linked to this article.

## Data availability

All data needed to evaluate the conclusions in this paper are present in the paper and/or the Supplementary Materials. Source data are provided with this paper.

## Code availability

The codes in MATLAB for the single-nanoparticle fluorescence quantification, fluorescence compensation, coincidence analysis, differentiation of different species, and deconvolution for payload quantification are available from the authors upon request.

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

## Acknowledgements

The authors would like to thank Dr. Robert K. Louder from Johns Hopkins University Integrated Imaging Center (IIC) for assistance in cryo-TEM experiments; Erin Pryce from IIC for assistance in Airyscan super-resolution imaging; and Professor Scot C. Kuo from the Institute for Basic Biomedical

Science of Johns Hopkins University School of Medicine for data analysis and interpretation in techniques relating to fluorescent imaging. This study was supported by the National Institute of Allergy and Infectious Diseases (NIAID) U01AI155313 to H.-Q.M. and R01AI137272 to T.-H.W.

## Author contributions

Y.H. conceptualized this study. S.L., Y.H., T.-H.W., and H.-Q.M. formed the study design and developed the major conclusions. S.L., A.L., and T.-H.W. developed the CICS platform and the numerical methods for resolving the payload characteristics. S.L., Y.H., and J.L. performed formulation experiments and data collection for CICS analysis of LNPs. J.L., Z.S., and E.K. contributed to the physical characterizations of the LNP formulations. K.H., P.Z., Y.Z., and C.Q. contributed to data analyses and scientific discussions. T.-H.W. and H.-Q.M obtained funding and supervised the study. The manuscript was written by S.L., Y.H., T.-H.W., and H.-Q.M., and all other authors contributed to the review and revision of the manuscript. T.-H.W., and H.-Q.M. contributed equally to this work as joint corresponding authors.

## Competing interests

S.L., Y.H., T.-H.W., and H.-Q.M. are co-inventors of a patent application covering the technique described in this paper, filed through and managed by Johns Hopkins University Office of Technology Ventures. The remaining authors declare no competing interests.
