## [Peer Review File · Nature Communications]

Payload Distribution and Capacity of mRNA Lipid NanoparticlesREVIEWER COMMENTS

Reviewer #1 (Remarks to the Author):

This manuscript describes studies of mRNA packaging in lipid nanoparticles, which is a timely and important subject. According to the authors, this study was possible primarily due to their single-molecule detection platform, which detects fluorescence from single particles as they flow through a capillary in 3 different fluorescence colors, thus allowing them to detect Cy-5 labeled mRNA, TMR-PC, and YOYO stained free mRNA. From the coincidence of the single-particle fluorescence signal, they then draw conclusion about whether and how much the lipid nanoparticles contain mRNA, helper-lipid, or whether the mRNA is unencapsulated. The study of the various parameters that might affect mRNA packaging was systematic and well performed and the results are clearly described, and the finding that 80% of the lipid nanoparticles do not contain mRNA is interesting and surprising, and certainly would be important for the field to know. However, there are several issues with the manuscript and statements made that should be addressed:

1) The mRNA has a very high content of Cy-5 (25%), which might affect the packaging. There needs to be some controls or experiments to show fluorescence labeling at such high levels do not affect mRNA packaging, and thus the conclusion made.

2) The manuscript seems to suggest that their single-molecule detection platform based on flowing sample through a capillary is what made this study possible and why the surprising finding that 80% of lipid nanoparticles (LNPs) were not known to the field before. In the introduction, it was stated that it was difficult to characterize payload distribution "primarily due to a technical gap for distinguishing empty LNPs from those with a payload". However, if the mRNA is labeled with Cy-5, why is it not possible to use fluorescence imaging, such as super-resolution imaging, to perform this study? It seems rather straightforward to do, and such instruments are also commercially available.

3) The conclusion that the mRNA was unencapsulated is based on YOYO staining and the absence of TMR lipid fluorescence. Is it possible that there is lipid present, but wasn't detected either because the percent of TMR lipid was low and so the lipid present didn't have TMR, or there were some TMR lipid but they were below the detection limit? There should be data showing single TMR-lipid detection.

4) The coincidence detection would be sensitive to fluorescence spillover as the authors indicated. The manuscript showed the spillover ratio, but doesn't show the error or uncertainty in this spillover ratio, which is the compensation spreading error. This should be determined.

Reviewer #2 (Remarks to the Author):

This a convincing and well-written manuscript. The results are important for the design of next-generation LNP mRNA systems. My only hesitation is that I would have liked to see the entrapment results (e.g. 80% of LNP mRNA formulations are "empty") verified by some orthogonal technique. There are density centrifugation techniques that should be able to provide equivalent information for certain special cases. I recommend that the authors perform one or two studies to show that the results obtained here are consistent with results obtained by other approaches.

Reviewer #3 (Remarks to the Author):

In this manuscript, the authors developed a single particle-analysis platform for the characterization of mRNA LNP formulations. Based on this platform, the authors characterized the payload distribution, capacity, the effects of PEG molar ratio in MC3 formulations at pH 4 and pH 7.4. Moreover, they hypothesized the assembly processes under different conditions. Overall, this is an interesting and

useful tool that may contribute to the development of mRNA LNP formulations. Here, I have several comments.

1. Table 3 shows “23% ± 8% mRNA-loaded LNPs; 77% ± 8% empty LNPs”. Is this based on the 3-color identification method?
2. The authors should explain the different encapsulation efficiency tested by RiboGreen and CICS.
3. For the result in Figure 5 k-m, LNP (N/P=6) showed obviously lower luminescence intensity in the spleen but not in the liver compared with LNP (N/P=3). Whether this difference may be attributed to the different biodistribution or in vivo stability of these two LNPs. Could the authors perform in vitro assays to compare the mRNA delivery efficiency and the stability between these two LNPs?
4. Optimization of LNP formulations to improve mRNA delivery efficiency is a potential application of this platform. Based on this platform, the authors obtained a lot of useful data about LNP parameters. It could be nice for the authors to provide some correlational analysis between LNP parameters and in vitro mRNA delivery efficiency.
5. The stability of mRNA LNP formulations in long-term storage is crucial for their clinical application. It would be meaningful if the authors provide the changes of LNP parameters (e.g., empty LNPs%, free mRNA%, average mRNA copy per LNP) after freeze and thawing circles.

Responses to Reviewer Comments for Manuscript Payload Distribution and Capacity of mRNA Lipid Nanoparticles

Reviewer #1

This manuscript describes studies of mRNA packaging in lipid nanoparticles, which is a timely and important subject. According to the authors, this study was possible primarily due to their single-molecule detection platform, which detects fluorescence from single particles as they flow through a capillary in 3 different fluorescence colors, thus allowing them to detect Cy-5 labeled mRNA, TMR-PC, and YOYO stained free mRNA. From the coincidence of the single-particle fluorescence signal, they then draw conclusion about whether and how much the lipid nanoparticles contain mRNA, helper-lipid, or whether the mRNA is unencapsulated. The study of the various parameters that might affect mRNA packaging was systematic and well performed and the results are clearly described, and the finding that 80% of the lipid nanoparticles do not contain mRNA is interesting and surprising, and certainly would be important for the field to know. However, there are several issues with the manuscript and statements made that should be addressed:

1) The mRNA has a very high content of Cy-5 (25%), which might affect the packaging. There needs to be some controls or experiments to show fluorescence labeling at such high levels do not affect mRNA packaging, and thus the conclusion made.

Response:

We completely understand the concern by the reviewer over the effect of Cy5 labeling on the package of mRNA molecules into the lipid nanoparticles, and would like to address it by the following:

- The Cy5-mRNA was synthesized by substitution of 25% of the uracil with Cy5-uracil during mRNA polymerization. Based on the sequence (<https://www.trilinkbiotech.com/cleancap-fluc-mrna.html>, under “CleanCap FLuc ORF Sequence” tab) provided by TriLink Biotechnologies, Inc., there are 235 uracil within this Cy5-mRNA. It is thus expected that on average of $235 \times 25\% = 59$ nucleotides are labeled with Cy5 on a single mRNA, contributing to only 3.1% of all the nucleotides in this 1929-nucleotide-long mRNA. From a statistical perspective, we expect the effect of Cy5 dye on the mRNA encapsulation in LNPs to be minimal.
- There have been numerous published studies in which this Cy5-mRNA product was used to replace non-labeled mRNA for investigation of the intracellular trafficking and biodistribution of mRNA LNPs. In these studies, the Cy5-mRNA-containing LNPs were used based on the assumption that they are equivalent to mRNA LNPs without a Cy5 label because of the statistically low labeling density. A few recent examples are listed below:
 - Dilliard SA, Cheng Q, Siegwart DJ. On the mechanism of tissue-specific mRNA delivery by selective organ targeting nanoparticles. *Proceedings of the National Academy of Sciences* (2021), 118(52) e2109256118.
 - Hajj KA, Melamed JR, Chaudhary N, Lamson NG, Ball RL, Yerneni SS, Whitehead KA. A potent branched-tail lipid nanoparticle enables multiplexed mRNA delivery and gene editing in vivo. *Nano Letters* (2020), 20(7) 5167-5175.

- Miao L, Li L, Huang Y, Delcassian D, Chahal J, Han J, Shi Y, Sadtler K, Gao W, Lin J, Doloff JC, Langer R, Anderson DG. Delivery of mRNA vaccines with heterocyclic lipids increases anti-tumor efficacy by STING-mediated immune cell activation. *Nature Biotechnology* (2019), 37, 1174-1185.
- We have manufactured the benchmark mRNA LNP formulation using non-labeled mRNA without TMR-PC and found that the z-average size, zeta-potential and encapsulation efficiency were very close to the those of LNPs manufactured using Cy5-labeled mRNA and TMR-PC. The comparison is shown below. This demonstrated that neither fluorescent labels significantly influenced the encapsulation and package behaviors of the LNPs. These data were added to the revised manuscript (**Fig. S4** and **Fig. S5**).

	Z-average diameter (nm)	Zeta-potential (mV)	Encapsulation efficiency
LNPs formulated with non-labeled mRNA and no labeled lipid	113.0 ± 1.4	-3.2 ± 1.4 mV	95.6% ± 0.6% by RiboGreen assay
LNPs formulated by Cy5-mRNA with TMR-lipid added	120.5 ± 6.0 nm	-6.3 ± 1.3 mV	94.2% ± 3.6% by RiboGreen assay

- We have carried out cryogenic transmission electron microscopy (Cryo-TEM) of the benchmark mRNA LNP formulation carrying Cy5-mRNA and TMR-PC fluorescent helper lipid and found that the morphology matched existing literature reports (*i.e.*, electron-dense spheres, as seen in References 5, 7 and 8). We added the images to **Fig. S1** to further enhance this point.

We hope that these justifications support the claim that the application of Cy5-labeled mRNA had minimal impact on the mRNA encapsulation and package behaviors in LNPs. This claim has been added to the first paragraph of the Methods section of the revised manuscript (Page 16, Lines 391–393).

2) The manuscript seems to suggest that their single-molecule detection platform based on flowing sample through a capillary is what made this study possible and why the surprising finding that 80% of lipid nanoparticles (LNPs) were not known to the field before. In the introduction, it was stated that it was difficult to characterize payload distribution "primarily due to a technical gap for distinguishing empty LNPs from those with a payload". However, if the mRNA is labeled with Cy-5, why is it not possible to use fluorescence imaging, such as super-resolution imaging, to perform this study? It seems rather straightforward to do, and such instruments are also commercially available.

Response:

We appreciate the comment from the reviewer that we need to better demonstrate the uniqueness of our technique. We first used the Zeiss LSM800 Airyscan super-resolution imaging modality from Johns Hopkins University Integrated Imaging Center. The modality has a resolution limit of 120 nm (*Nat Methods* 12, i–ii, 2015. <https://doi.org/10.1038/nmeth.f.388>), which approaches the size of the mRNA LNPs. With this resolution, the presence of empty LNPs was resolved by observing the benchmark formulation embedded in solidified resin (ProLong Diamond Antifade Mountant from Invitrogen), shown below. The white arrows in the zoom-in area 1 and 2 showcase presence of a large quantity of empty LNPs (TMR⁺

Cy5⁻), while the yellow arrows in the zoom-in area 1 showcase presence of unencapsulated mRNAs (TMR⁻ Cy5⁺). This figure has been added to Supporting Information as **Fig. S11** in the revised manuscript.

However, we feel not confident in extracting quantitative data out of these images because of the small sample size, and more importantly, that super-resolution imaging modality (like the Airyscan we used) is typically equipped with gallium arsenide phosphide photomultiplier tube (GaAsP-PMT) detectors. These detectors have lower signal-to-noise ratios compared to the single-photon counting avalanche photodiodes (APD) detectors that we used in our study, especially in assessing nanoparticles with a low fluorescent intensity and unpackaged mRNA. Our analysis (main text **Fig. 2g**) clearly showed that the average TMR signal intensity of empty LNPs was much lower than that of mRNA-loaded LNPs. This was somewhat captured by these confocal images that the empty LNPs appeared to be dimmer in TMR channel compared to mRNA-loaded LNPs. Therefore we feel that it is important to note that **there was a possibility that some empty LNPs were missed by this super-resolution imaging.**

The other important aspect of our manuscript, the mRNA payload capacity (number of mRNA copies per nanoparticle), which requires precise fluorescent quantifications, cannot be obtained by these super-resolution images. It shares the same limitations of other fluorescence microscopic detection including the intrinsic variation of the number of fluorophores labeled per molecule, fluctuations and photobleaching of fluorescence during detection, etc., which make it very difficult to isolate and study the effect of number of

mRNA encapsulated that contribute to the detected fluorescent signal. Our CICS method utilizes the fluorescent distribution of free mRNA to deconvolute that of mRNA LNP formulations to obtain the single nanoparticle payload information. Since the aforementioned variations are contained and presented in the fluorescent distributions, the deconvolution analysis is able to isolate the number of mRNA encapsulated for accurate description of the payload distribution. These claims are supported by the references 22 and 23 in the manuscript, and we briefly discussed this in the revised manuscript (Page 6, Lines 125–134).

Our flow-based spectroscopic method also holds several other advantages compared to the fluorescent imaging-based methods. For example:

- Preservation of the natural solution condition of the lipid nanoparticle samples: Imaging methods usually require physical embedding that may alter the physical and chemical properties of nanoparticles, while the experiments described by our manuscript were carried out under the original buffer condition of the mRNA LNPs.
- Higher sample screening throughput: High-resolution imaging usually require prolonged (minutes to hours) scanning time with limited number of nanoparticles (hundreds) analyzed, while our method can assess 3000 to 5000 particles per minute. Our quantitative assessment was typically based on an ensemble of 10^4 – 10^5 particles for each analysis.
- Automation and higher degree of instrument control: Our method only requires sample loading into a pressure chamber with minimal user input during the measurements, and each experiment only takes less than 20 minutes to run.

3) The conclusion that the mRNA was uncapsulated is based on YOYO staining and the absence of TMR lipid fluorescence. Is it possible that there is lipid present, but wasn't detected either because the percent of TMR lipid was low and so the lipid present didn't have TMR, or there were some TMR lipid but they were below the detection limit? There should be data showing single TMR-lipid detection.

Response:

We thank the reviewer for raising this noteworthy question. The TMR-PC helper lipid was dosed at 0.5 mol% of all lipid components, thus 5 mol% of the helper lipids in the mRNA LNP formulations characterized. We carefully chose this ratio because in preliminary experiments we found that a TMR-PC ratio greater than 1.0 mol% of all lipids would significantly reduce the encapsulation efficiency and increase the LNP size. As suggested by the reviewer, we carried out an analysis of a pure TMR-PC solution to resolve the single-TMR-PC signal spectrum, shown below.

The fluorescent intensities of single TMR-PC signal were close to our detection threshold of 10 photons within this channel because each TMR-PC molecule only contains a single TMR fluorophore. Besides, the fact that this fluorescent helper lipid was added at a 5 mol% of the helper lipid means that the distribution of TMR fluorophore in LNPs follows a binominal distribution, further complicating the data interpretations from the TMR intensity profiles. Therefore, we do not think it is accurate to obtain any strict quantifications out of the TMR channel, and we framed the message throughout the manuscript that the analysis of TMR signal is only relative as an indicator of relative helper lipid content.

We fully agree with the reviewer that it is possible that some events, either YOYO-1-bound mRNAs, non-lipophilic complexes specifically defined in pH 4.0, and empty LNPs, may contain trace amounts of helper lipid that was not captured by TMR detection. However, we do not think this will negatively impact the major conclusions of our manuscript:

- As shown in the figure above and in **Fig. 2g**, mRNA-loaded and empty LNPs have drastically different TMR signal intensity thus different helper lipid content. While some TMR signals out of the empty LNPs approach single-TMR level, we anticipate a certain degree of underestimation of the fraction of empty LNPs. This does not influence our conclusion and discussions about the fact that a significant portion of the LNPs were not loaded with an mRNA.
- YOYO-1-bound mRNAs with a positive TMR signal only contributed to a tiny portion of all events (<0.1% of total events). We believe those were staining artifacts which were excluded from downstream quantitative analysis by coincidence analysis.
- The non-lipophilic complexes have significantly lower TMR signal intensity compared with lipophilic complexes under pH 4.0, and the difference is substantial to make a solid distinction between the two populations. We agree that these non-lipophilic complexes may contain trace amounts of helper lipid, so we softened our descriptions in the revised manuscript:
 - “...suggesting they might be non-lipophilic, highly charged complexes of mRNA and ionizable lipids that could not accommodate as many helper lipids did not favor helper lipid insertion.”

4) The coincidence detection would be sensitive to fluorescence spillover as the authors indicated. The manuscript showed the spillover ratio, but doesn't show the error or uncertainty in this spillover ratio, which is the compensation spreading error. This should be determined.

Response:

We appreciate this comment from the reviewer as it is indeed important to show the spillover spreading along with the compensation ratio. As suggested, we have now added the spillover spreading matrix (SSM, shown below) based on the definition by Nguyen et al. in their work “Quantifying Spillover Spreading for Comparing Instrument Performance and Aiding in Multicolor Panel Design” (Cytometry Part A, 2013, DOI: 10.1002/cyto.a.22251). The off-diagonal elements are the intrinsic spillover spreading value that indicate how much error is contributed to the detectors by the fluorophores. This table has been added as **Table S1** in the Supporting Information of the revised manuscript.

Table S1. Spillover spreading matrix (SSM) of the 3-color CICS experiments

		Detector		
		520/28	575/25	676/37
Fluorophore	YOYO-1	N/A	0.7909	0
	TMR	0.648	N/A	1.0132
	Cy5	0	0.9368	N/A

Reviewer #2

This a convincing and well-written manuscript. The results are important for the design of next-generation LNP mRNA systems. My only hesitation is that I would have liked to see the entrapment results (e.g. 80% of LNP mRNA formulations are "empty") verified by some orthogonal technique. There are density centrifugation techniques that should be able to provide equivalent information for certain special cases. I recommend that the authors perform one or two studies to show that the results obtained here are consistent with results obtained by other approaches.

Response:

We first would like to thank the reviewer for the positive feedbacks.

With regard to the suggestion of using analytical ultracentrifugation (AUC) as an orthogonal verification method to characterize the mRNA LNPs, we held discussions with AUC experts from the Department of Biophysics at Johns Hopkins University but were advised against it due to extreme complicity and cost associated with setting up such experiments. There has been a published report using AUC to analyze payload features of LNPs in 2021 titled “Density matching multi-wavelength analytical ultracentrifugation to measure drug loading of lipid nanoparticle formulations” (DOI: 10.1021/acsnano.0c10069, the reference 12 in our manuscript). In this work, Professor Borries Demeler *et al.* developed an AUC-sedimentation velocity experiment (SVE) methodology and a data processing algorithm to evaluate siRNA LNPs. To obtain the sedimentation coefficient distributions of LNP formulations, a density matching strategy using a series of D₂O/H₂O mixture as buffer was used. Payload features were then obtained through derived partial specific volume distributions. This published study found that there were no empty LNPs, and we have discussed this difference in our manuscript in the section “Fraction of empty LNPs” (Page 13, Lines 310–316). We think the nucleic acid cargo size plays a critical role, for which a smaller size (meaning a higher molar concentration) favors more uniform lipid precipitation onto the nucleic acids instead of self-precipitation, thus reducing the fraction of empty LNPs. This agrees with our data when a shorter mRNA was used (**Fig. 5j**). At this moment, we are carrying out experiments with siRNA LNPs to cross-verify the findings with this publication, and we hope to report our findings in a future manuscript.

Inspired by the second comment from Reviewer #1, we used a super-resolution fluorescent imaging technique (Airyscan by Zeiss as available at Hopkins) to directly observe the mRNA LNP formulation embedded in solidified resin (ProLong Diamond Antifade Mountant from Invitrogen). The theoretical resolution of this imaging modality is 120 nm, approaching the size of the LNPs. Without the capability to do any quantification, it was though able to show distinguishment of empty vs. loaded LNPs. As we attach the images here on the next page, the white arrows in the zoom-in area 1 and 2 indicate the presence of a large quantity of empty LNPs (TMR⁺ Cy5⁻), while the yellow arrows in the zoom-in area 1 showcase the presence of a few unencapsulated mRNAs (TMR⁻ Cy5⁺). Our analysis (main text **Fig. 2g**) clearly shows that the average TMR signal intensity of empty LNPs was much lower than that of mRNA-loaded LNPs. This was somewhat captured by these confocal images that the empty LNPs appeared to be dimmer in TMR channel compared to mRNA-loaded LNPs. Therefor we feel that it is important to note that **there is a possibility that some empty LNPs were missed by this super-resolution imaging**. In other words, we expect underestimation of the number of empty LNPs by Airyscan. We feel that this is a strong evidence showcasing the presence of empty LNPs as an important conclusion of our manuscript. This figure has been added to Supporting Information as **Fig. S11** in the revised manuscript.

For other important conclusions of our manuscript, such as the payload capacity (mRNA and lipid content per LNP) and payload behaviors along with the manufacturing processes, we hope to seek for additional verification methods in the future.

Reviewer #3

In this manuscript, the authors developed a single particle-analysis platform for the characterization of mRNA LNP formulations. Based on this platform, the authors characterized the payload distribution, capacity, the effects of PEG molar ratio in MC3 formulations at pH 4 and pH 7.4. Moreover, they hypothesized the assembly processes under different conditions. Overall, this is an interesting and useful tool that may contribute to the development of mRNA LNP formulations. Here, I have several comments.

1. Table 3 shows “23% ± 8% mRNA-loaded LNPs; 77% ± 8% empty LNPs”. Is this based on the 3-color identification method?

Response:

Yes, these results were obtained from the 3-color identification method. We have added description to clarify this point in the main text introducing **Table 3** (Page 8, Lines 187).

2. The authors should explain the different encapsulation efficiency tested by RiboGreen and CICS.

Response:

We thank the reviewer for pointing this out.

Firstly, after carefully reviewing the data and calculations of the encapsulation efficiency (EE%) data shown in **Fig. S5** in our manuscript, we found that the CICS-derived EE% of N/P = 12 formulation had an error that made it very different from the RiboGreen-derived EE%. This has been corrected in the revised manuscript. Now for this formulation, CICS and RiboGreen reported comparable results, 89.7% and 94.8%, respectively.

For most of the formulations, CICS showed only slight differences compared with RiboGreen. We attribute these differences to intrinsic deviations of both methods. RiboGreen is a colorimetric method based on comparison of the average or collective binding degree of the dye to free mRNA molecules vs. all mRNA molecules released by LNP disruption. The major source of deviation is uncertainty of dye penetration into LNPs and dye binding to lipid components that will trigger fluorescence emissions, which cannot be calibrated by standard curves. Our CICS method used fluorescence coincidence analysis to identify YOYO-1⁻, TMR⁺ and Cy5⁺ events first (main text **Table 2**), and then use the absolute mRNA amount quantified in these mRNA-loaded LNPs to calculate EE% through dividing it by the total mRNA fluorescence quantified (described in **Supporting Discussion**). The biggest origin of deviation lies with a certain degree of uncertainty of YOYO-1 penetration and TMR thresholding level to define TMR⁺.

The only outlier was the formulation of N/P = 1, for which CICS and RiboGreen showed drastically different results. This formulation is very special because it is anticipated that the mRNA encapsulation would be poor as the ionizable lipids are insufficient to neutralize all the negative charges on mRNA molecules. We hypothesize that there would still be certain degrees of lipid complexation onto the mRNA molecules, however, the complexation could not fully protect the mRNAs from binding to RiboGreen dye. RiboGreen therefore gave a very low EE% reading of 27.8%. However, the same degree of lipid complexation might accommodate enough TMR-PC lipids to make its signal exceed the detection threshold; besides, the YOYO-1 dye we used might not penetrate to bind to mRNAs as well as RiboGreen. We found that in this case most of the Cy5 events were still recognized as mRNA-loaded LNPs by CICS and it gave a high EE% reading of 82.9%. This difference should not be recognized as an error, but it represents

different aspects of the encapsulation of mRNA under different N/P ratio (or compaction degree) conditions. CICS and RiboGreen gave very similar EE% results in all conditions other than this special case of N/P = 1. We are confident that our technique is able to assess this parameter accurately.

We added the above analysis to the section 4 to describe the CICS EE% method in the **Supporting Discussion** of the revised manuscript.

3. For the result in Figure 5 k-m, LNP (N/P=6) showed obviously lower luminescence intensity in the spleen but not in the liver compared with LNP (N/P=3). Whether this difference may be attributed to the different biodistribution or *in vivo* stability of these two LNPs. Could the authors perform *in vitro* assays to compare the mRNA delivery efficiency and the stability between these two LNPs?

Response:

With regard to the suggestion from the reviewer to test *in vitro* transfection efficiency, we hesitated to do so because of the transfection of LNPs in the lungs, liver and spleen was shown to be strongly correlated with interactions with blood serum components. It was demonstrated by reference 27, 28 cited in our manuscript for liver, and one recent publication for lungs and spleen (Siegwart et al., 2021, *PNAS*, DOI: 10.1073/pnas.2109256118). Therefore, we think that *in vitro* experiments would only lead to weak, less informative conclusions. However, we gladly accepted the suggestion from the reviewer to carry out a biodistribution experiment of the three formulations we tested *in vivo*. The results are shown as **Fig. S12** in the revised manuscript and attached below. We added relevant descriptions and explanations both in the main text (Page 14, Lines 328–331) and below this supplementary figure in the revised manuscript.

Figure S12. Biodistribution of mRNA LNP formulations tested *in vivo*. In this experiment, the same animal experimental procedures as described in the **Methods** section in the main text were used with several modifications: (1) The cargo was Cy5-mRNA for the purpose of tracking the biodistribution of the mRNA-loaded LNPs; (2) The IVIS live-animal imaging was performed to harvested organs at 4 hours post-injection with fluorescence detection mode to Cy5; (3) The homogenized organ solution samples were analyzed by a plate reader in detection of Cy5. The results are shown in (a) and (c) for liver, and in (b) and (d) for spleen. For statistically analysis in (d) an unpaired t test was performed with ** denoting p (two sided) < 0.01 . In (a) and (b), the scale represents fluorescence radiant efficiency with the unit of $(\text{p/sec/cm}^2/\text{sr})/(\mu\text{W/cm}^2)$.

The results showed that the biodistribution of the mRNA LNPs to the liver did not differ significantly across different groups, meaning that the decrease of transfection efficiency in the liver caused by addition of extra empty LNPs into the $N/P = 3$ formulation could not be explained by different biodistribution profiles. The biodistribution to spleen was significantly lower for $N/P = 6$ compared to the other two formulations based on $N/P = 3$, and it correlates with the lower transfection efficiency of $N/P = 6$ formulation in the spleen. However, we could not conclude that the biodistribution profile was the only reason that led to the transfection profiles.

4. Optimization of LNP formulations to improve mRNA delivery efficiency is a potential application of this platform. Based on this platform, the authors obtained a lot of useful data about LNP parameters. It could be nice for the authors to provide some correlational analysis between LNP parameters and *in vitro* mRNA delivery efficiency.

Response:

We fully agree with the reviewer that the effects of payload distribution and capacity of mRNA LNPs are of great interest to the research and industrial communities. However, we feel reluctant to do *in vitro* screening experiments because at this stage we cannot independently control payload distribution and capacity. For example, by changing the dosage of PEG lipid (Fig. 3), payload capacity can be changed, however, it is coupled with different sizes of the LNPs. While size was known to significantly impact transfection efficiency (e.g., reference 15 in our manuscript), it would be difficult to get strong, reliable conclusions out of the experiments. By changing the N/P ratio (Fig. 4 of the manuscript), payload capacity can be changed while controlling the same LNP size, however, it is coupled with different overall lipid dose, which is a dominant factor influencing transfection efficiency. Besides, we currently do not have a method to control the fraction of empty LNPs other than adding separately formulated empty LNPs into existing formulations (Fig. 5), leaving us only a tight parameter space.

Therefore, we feel that even such experiments can be done to all the formulation conditions we have, the results would be based on too many assumptions, and the implications and conclusions would be weak and can only offer marginal supplementation to the body of the work of this manuscript. We hope the reviewer can understand our judgements. Again, we would like to be conservative and cautious in making any conclusions over the biological activities of this benchmark formulation, while providing as much information on the assembly side as possible.

5. The stability of mRNA LNP formulations in long-term storage is crucial for their clinical application. It would be meaningful if the authors provide the changes of LNP parameters (e.g., empty LNPs%, free mRNA%, average mRNA copy per LNP) after freeze and thawing cycles.

Response:

We thank the reviewer for this suggestion. We performed this experiment and reported our findings in **Fig. S13** (also attached here) in the revised manuscript. We also added relevant description referencing to this data in the main text. The following discussions were added to **Fig. S13**.

Figure S13. Monitoring payload distribution and capacity along with the storage of mRNA LNPs under different conditions. In this experiment, the LNPs were formulated with 5% sucrose as a validated single cryo-protectant. Upon formulation to pH 7.4, the samples were stored either in 4°C, -20°C, or -80°C, which are common storage conditions for mRNA LNPs. At 5 days, 11 days and 36 days post-storage, the samples were warmed up to room temperature and analyzed by CICS. The (a) z-average size; (b) average mRNA copy per nanoparticle among mRNA-loaded LNPs; (e) geometric mean of TMR signals (indicator of relative helper lipid content) of mRNA-loaded LNPs, (f) geometric mean of TMR signals of empty LNPs, and (g) fraction of empty LNPs are shown. The payload distribution profiles of the samples assessed at (c) 5 days, and (d) 36 days post-storage, respectively, are also shown to illustrate the payload distribution changes.

While we were unable to perform this experiment with the formulation as in the real mRNA LNP product on the market, an existing publication has validated 5% sucrose as an effective single cryo-protectant for long-term low-temperature storage⁴. Under this storage condition, a series of changes in the payload distribution and capacity of the mRNA LNP formulation was observed. Firstly, no significant differences were observed between the samples stored under -20°C and those under -80°C (**Fig. S13b–g**) except -20°C resulted in greater size increase upon thawing (**Fig. S13a**). The samples stored under 4°C did not show appreciable quality drop over the entire storage period of 36 days. While

numerous reports (including the paper we reference to⁴) clearly demonstrated that the biological activity of mRNA LNP formulations quickly decreased to zero a few days upon stored under 4°C, we showed through our experiments that it was not due to collapse of the LNP structure or assembly features. This agrees with literature report that hydrolysis of the mRNA molecules might be the major culprit for instability of mRNA LNP formulations under elevated temperature⁵. Even after mRNA molecules are broken down to shorter, unfunctional strands, they are still complexed in the LNP core, leaving payload distribution and capacity of the LNPs unchanged. We also observed a trend of increasing helper lipid content in the mRNA-loaded LNPs, but the underlying mechanisms are currently unclear to us, for which we will continue to investigate in our future studies.

REVIEWER COMMENTS

Reviewer #1 (Remarks to the Author):

The authors' effort to address the reviewers' comments is appreciated and the manuscript has improved. However, there are still some issues or points the authors should consider:

1) The authors have provided the spillover spreading, and in looking at FigS3, it seems spillover ratio can vary by several fold in the range of a typical TMR signal, for example. Figure S3e shows at 100 photons, the ratio can vary from 0.05 to 0.4, which is 8 times. In h, after compensation, a large fraction of the data points has negative photon burst size (less than 0). Since the conclusion about 80% empty particles rely on looking at coincidence of the dyes after spillover correction, how does this seemingly significant spillover spread translates into error in the percentage of empty particles?

2) It is stated in the paper on page 16 that "we verified the presence of fluorescent tags did not alter the size, surface charge,". But the table in the response letter shows the magnitude of zeta potential doubled, which implies the surface charge doubled. If so, the fluorescent tags did change the surface charge, contrary to the statement made in the paper. Some clarifications here would be helpful.

3) The authors performed airyscan confocal microscopy to get higher resolution and stated the imaging method is unreliable for quantitative determination. The manuscript also maintains fluorescence microscopy cannot elucidate the results presented in the paper. However, the authors did not use the right type of microscopy because confocal is often not suitable for attaining single molecule sensitivity as it is prone to bleaching the dyes. There are other modes of microscopy (such as STORM, TIRF) that can achieve single molecule sensitivity and it seems those methods (quite routine now) would be able to provide the same level of quantitative information.

4) The conclusion that 80% of particles using the benchmark formulation is empty seems to suggest there isn't enough mRNA and there are excess lipids. The authors varied the concentration of mRNA to lipids but kept the mass ratio of lipid to mRNA the same. Wouldn't simply increasing the amount of mRNA while keeping all lipid composition the same decrease the amounts of empty particles? Would the unencapsulated mRNA increase significantly from the 4%?

Reviewer #2 (Remarks to the Author):

The authors have provided detailed responses to reviewer's comments, however they have not performed a definitive experiment to verify their results using an independent technique. Thus I am not entirely convinced that the fundamental finding that 80% of the LNP are "empty" is correct. However, I do find it highly likely that a substantial portion are. This manuscript will stimulate further investigations in what is clearly and important parameter affecting the potency of LNP mRNA formulations and I think warrants publication.

Reviewer #3 (Remarks to the Author):

Thanks for the authors' work and responses. I have no further questions.

Responses to Additional Reviewer Comments for Manuscript Payload Distribution and Capacity of mRNA Lipid Nanoparticles

Reviewer #1

The authors' effort to address the reviewers' comments is appreciated and the manuscript has improved. However, there are still some issues or points the authors should consider:

1) The authors have provided the spillover spreading, and in looking at Fig S3, it seems spillover ratio can vary by several fold in the range of a typical TMR signal, for example. Figure S3e shows at 100 photons, the ratio can vary from 0.05 to 0.4, which is 8 times. In h, after compensation, a large fraction of the data points has negative photon burst size (less than 0). Since the conclusion about 80% empty particles rely on looking at coincidence of the dyes after spillover correction, how does this seemingly significant spillover spread translates into error in the percentage of empty particles?

Responses: We appreciate this comment from the reviewer prompting us to carefully examine the reliability of the data and conclusions presented in our manuscript. In particular, signal bleeding and compensation into TMR and Cy5 may have influence in data processing to derive empty LNP ratio and mRNA payload in LNPs. We would like to address this point from the following perspectives:

- Even though we have showcased the compensation of signals in our manuscript using samples purposely having high single fluorescent signals, we found that only a small portion of the realistic signals from these samples resulted in a signal intensity that exceeded the detection thresholds in the “bleeding into” channels. For the single stained sample (shown below in **Table R1**), the percentage of the signals that bleed into the neighboring channels are only a few percent except the one from YOYO-1 to TMR. The 27.2% of the signal spillover from YOYO-1 to TMR in the single stained control experiment is due to the large DNA molecules we used (Hind III digested lambda) which contains long strands with extremely high signal intensity. This did not apply to the actual sample as free mRNAs stained by YOYO-1 would yield generally lower level of signals. For TMR single stained sample, only 2.6% of the signals caused bleeding, and the LNP formulations of interest would only have lower, and fewer such “significant signals” that needed compensation.

Table R1. The frequencies of bleed-through events using single stained samples

		% Signal bleeding into		
		YOYO-1	TMR	Cy5
From	DNA-YOYO-1	N/A	27.2%	0.2%
	mRNA LNP, 0.25% PEG TMR only	1.2%	N/A	1.5%
	mRNA LNP, 0.25% PEG Cy5 only	1.2%	2.6%	N/A

- Therefore, signals from LNP formulations of interest that we presented in the manuscript are predominantly of a reasonably compensation-insensitive intensity. This was a result of the intense optimizations that we performed at the beginning of this study as we carefully selected and optimized the laser sources, filters, fluorophores, and TMR-PC blending ratios to work as much as

we can in a region where good signal-to-noise ratio can be maintained while minimizing the spillover ratio. This served as the most critical factor that contributed to the accurate quantitative assessments as described in our manuscript.

- The empty LNP assessments would be affected the most by the TMR bleed through into Cy5 channel or the other way, which will be considered as false-positive mRNA-loaded LNP events. Note that YOYO-1 bleeding into TMR does not pose any effect because YOYO-1 containing events are immediately excluded from LNP analysis. Here we chose two conditions (0.5% PEG lipid, and N/P = 2 at pH 7.4) as examples: 0.5% PEG lipid yielded one of the highest LNP TMR signal (**Fig. 3f**), and N/P=2 yielded one of the highest Cy5 signal (**Fig. 4e**) with the highest likelihood of bleeding. In addition, both conditions gave one of the highest empty LNP percentages (**Figs. 3h** and **4h**). To estimate the error caused by the fluorescent compensation, we compared two of the key parameters reported by CICS (*i.e.*, mRNA payload per loaded LNP and empty LNP percentage) that were obtained with or without compensation. **Table R2** below shows that the compensation only minimally affects the empty particle percentage and mRNA payload quantification.

Table R2. Comparison of the empty LNP percentage and average mRNA per loaded LNP with or without fluorescence compensation for two representative LNP formulations

Condition	Replicate	Empty LNP %			Average mRNA per LNP		
		w/ compensation	w/o compensation	Difference	w/ compensation	w/o compensation	Difference
0.5% PEG lipid	#1	76.9%	76.8%	0.03%	5.19	5.13	0.06
	#2	85.3%	85.2%	0.04%	7.30	7.31	-0.01
	#3	76.3%	76.2%	0.06%	8.18	8.12	0.06
N/P=2	#1	82.4%	82.4%	0.01%	6.86	6.78	0.08
	#2	83.7%	83.7%	0.01%	7.36	7.36	0.00
	#3	71.9%	71.9%	0.01%	11.01	10.95	0.06

- With regards to the reviewer’s note on widely spread spillover ratios, it is not a specific issue for CICS but rather for any multi-color fluorescent detection system, such as flow cytometry set-ups. Fixed ratio strategy cannot prevent imperfect compensation in which some signals will be over-compensated, while some others will be under-compensated. Therefore, we totally understand the reviewer’s concern about quantifications with less ideal spread of spillover / compensation ratios.
- We have demonstrated that our current work in this present manuscript does not rely on accurate compensation to obtain accurate results. However, we would like to keep this part in the manuscript still as an essential piece of data. Similarly as flow cytometry in assessing cells, for which compensation is a routine step, we would like to show the feasibility of fluorescent compensation for CICS platform in assessing nanoparticles.

We hope this helps to address the concern over the accuracy of our analytical method.

2) It is stated in the paper on page 16 that "we verified the presence of fluorescent tags did not alter the size, surface charge,". But the table in the response letter shows the magnitude of zeta potential doubled, which implies the surface charge doubled. If so, the fluorescent tags did change the surface charge, contrary to the statement made in the paper. Some clarifications here would be helpful.

Responses: We thank the reviewer for pointing this out. We indeed noticed the zeta-potential difference between LNPs prepared from fluorescently labeled vs. unlabeled components. However, the difference, even nearly 2 folds (-3.2 ± 1.4 mV for LNPs formulated with non-labeled mRNA and no labeled lipid vs. -6.3 ± 1.3 mV for LNPs formulated with Cy5-mRNA with TMR-lipid added), is considered not substantial due to the fact that the particle surface is considered near electrostatically neutral (when zeta potential is within $\pm 0 - 10$ mV). At near neutral condition, value fluctuations are more pronounced. Therefore, this level of difference for near-neutral surface charged nanoparticles does not indicate physically meaningful differences in the charge properties of the LNPs.

It is important to point out that zeta-potential measurement as a method to estimate surface charge characteristics of nanoparticles is rather a semi-quantitative method. A reference we would like to cite is the review by Prof. Sourav Bhattacharjee in *Journal of Controlled Release* in 2016 titled "DLS and zeta potential – What they are and what they are not" (10.1016/j.jconrel.2016.06.017).

"It should be noted that ZP [Note: zeta potential] never measures charge or charge density and rather deals with surface potential. Therefore, only the magnitude of ZP is important while the positive/negative finding associated with it is not robust and should not be related with surface charge or charge density or making comparisons between different nanoformulations. ZP only provides with indicative evidence towards the nature of surface charge (positive/negative) assuming that the predominant ions in the EDL [Note: electric double layer] up to the slipping plane are similar (positive/negative) compared to the surface of the particle itself."

In nanoformulation systems in which surface charge repulsion is the only force for stabilization, practical categorization is also based on magnitudes of zeta-potential, but not absolute measures of it, giving a general sense of why a zeta-potential near neutral can have definitive and non-biased interpretations even with larger-than-usual differences in values. The review article states that:

"Guidelines classifying NP-dispersions with ZP values of $\pm 0-10$ mV, $\pm 10-20$ mV, $\pm 20-30$ mV and $> \pm 30$ mV as highly unstable, relatively stable, moderately stable and highly stable, respectively are common in drug delivery literature."

We therefore consider the values of -3.2 ± 1.4 mV (for LNPs formulated with non-labeled mRNA and no labeled lipid) and -6.3 ± 1.3 mV (for LNPs formulated with Cy5-mRNA with TMR-lipid added) both as near-neutral-to-weakly negative zeta potential. This range of zeta potential values agrees well with many literature reports of LNPs formulated with the same lipid components and the same compositions as those used in our study. Our reported zeta potential values fall within the range of $+0.9$ mV to -8 mV as shown in **Table R3** below.

Table R3. The typical range of zeta potential values reported for the LNPs formulated with the same lipid components and compositions as those used in our study

DOI	Citation	Data or statements
10.1038/s41467-018-06936-1	Nature Communications (2018) 9, 4493.	Supplementary Table 1: +0.9 ± 0.28 mV and +0.5 ± 0.42 mV Note: even with a positive number, it still indicates a near neutral surface charge
10.1021/acs.nanolett.8b01101	Nano Letters (2018) 18, 3814-3822	Supplementary Table Ranges from -2.91 ± 0.54 mV to +0.65 ± 0.52 mV
10.1038/s42003-021-02441-2	Communications Biology (2021) 4, 1-15.	“With a pKa near 6.5, the LNP at neutral pH is negatively charged (from the nucleic acid payload).” Figure 1c: Around -8 mV based on the DLin titration curve
10.1016/j.xphs.2021.11.004	Journal of Pharmaceutical Sciences (2021) 111, 690-698	Figure 1c: Around -5 mV with an error bar of around ±2 mV

To improve accuracy of the description, we modified the relevant discussion as shown below. Prompted by the reviewer comments, we felt the importance to clearly state the (non-substantial) effects from the fluorescent labels as we believe the results presented in our manuscript based on characterization of fluorescent LNPs are representative for normal LNPs. We have thus moved the statement below into the main text of this manuscript, at the end of the first paragraph of **Results**, and supplemented a table (**Supplementary Table 1**) to show the comparisons and reasonings above.

“We verified that the presence of fluorescent tags (on Cy5-mRNA and TMR-PC) did not **perceptibly** alter the size, **the near-neutral nature of surface charge**, the encapsulation efficiency (**Supplementary Table 1**) or the morphology (**Supplementary Fig. 1**) of the mRNA LNPs.”

3) The authors performed airyscan confocal microscopy to get higher resolution and stated the imaging method is unreliable for quantitative determination. The manuscript also maintains fluorescence microscopy cannot elucidate the results presented in the paper. However, the authors did not use the right type of microscopy because confocal is often not suitable for attaining single molecule sensitivity as it is prone to bleaching the dyes. There are other modes of microscopy (such as STORM, TIRF) that can achieve single molecule sensitivity and it seems those methods (quite routine now) would be able to provide the same level of quantitative information.

- **Related comments in the first round of revision:** The manuscript seems to suggest that their single-molecule detection platform based on flowing sample through a capillary is what made this study possible and why the surprising finding that 80% of lipid nanoparticles (LNPs) were not known to the field before. In the introduction, it was stated that it was difficult to characterize payload distribution "primarily due to a technical gap for distinguishing empty LNPs from those with a payload". However, if the mRNA is labeled with Cy-5, why is it not possible to use fluorescence imaging, such as super-resolution imaging, to perform this study? It seems rather straightforward to do, and such instruments are also commercially available.

Responses:

We agree with the reviewer and have consulted with Dr. Scot C. Kuo, Director of the Basic Sciences Microscope Facility and Associate Professor of Biomedical Engineering & Cell Biology at Johns Hopkins University School of Medicine, for expert opinion on the high-performance imaging techniques.

We realized that in our last responses to the reviewer comments, we falsely overclaimed that the mRNA payload capacity (mRNA copies per LNP) cannot be obtained by super-resolution imaging techniques, such as single-molecule TIRF. Our descriptions in **Introduction** were arbitrary in stating "*It has been difficult to characterize the payload distribution and capacity in these LNPs at a single-nanoparticle level, primarily due to a technical gap for distinguishing empty LNPs from those with a payload...*" In principle, these fluorescent imaging techniques are capable to obtain such information through the same deconvolution analysis that we used by comparing fluorescent signals from free mRNA vs. mRNA-loaded LNPs. However, validating these quantitative techniques for the delicate LNP system is non-trivial and beyond the scope of this study. Some practical considerations include:

- **Diffusion:** Particle diffusion, particularly across the thin evanescent range (~100 nm) of TIRF microscopy, not only blurs the obtained images, but also affects particle brightness. Quantitatively, diffusion would further broaden brightness distributions and greatly complicate the deconvolution analysis approach. Although particle-immobilization approaches exist, optimizing the immobilization method and validating the integrity of the LNPs requires extensive effort and warrants its own study, as our results indicate the lipid structures can be fluidic with compositional drift as solution environment changes, so immobilizing particles (using multivalent salts, covalent bonds, etc.) risks perturbing the structural integrity of LNPs.
- **Random labelling & chemical inaccessibility:** Single-molecule localization microscopy (SMLM) techniques, most notably direct stochastic optical reconstruction microscopy (dSTORM) and photoactivated localization microscopy (PALM), can be used to quantify the fluorescence of the mRNA within LNP. Not only must the LNPs be completely immobilized for these slow imaging techniques (see above), the labeling nature of the mRNA (25% of Cy5-uridine) prevents direct interpretation of imaging data. Due to the long contour length of mRNA (1929 nt in our case), the randomly labeled mRNA molecules are likely to be entangled, condensed, and ultimately unresolvable as separate molecules. Furthermore, specialized buffers needed to scavenge oxygen and promote fluorophore dark-state (dSTORM) may not penetrate the LNP shell as they cannot

penetrate cells. Although alternative approaches exist, the delicacy of LNPs means exploration is a dedicated effort beyond the scope of this study.

We agree with the reviewer that high-performance imaging is a robust tool capable of studying the payload distribution and capacity, and potentially microscopic structures of LNPs. Technical challenges prevent our application here, and present significant, but not impossible hurdles. We hope our study can inspire such efforts in the future, and we therefore modified our statements in **Introduction**, and added the following forward-looking descriptions in the **Conclusion** section of the revised manuscript, shown below:

Introduction

“The payload distribution and capacity of mRNA LNPs are important characteristics to assess, because they hint at molecular assembly mechanisms and influence pharmacodynamics, pharmacokinetics, and delivery efficiency^{7, 12, 13}. However, Cryo-TEM, and other common nanoparticle characterization methods such as small-angle neutron scattering⁷, NMR¹⁴, and nanoparticle tracking analysis¹⁵, could not effectively resolve the payload distribution and capacity of mRNA LNPs at a single-nanoparticle level, primarily due to difficulty in distinguishing empty LNPs from those with a payload^{9, 16} (Supplementary Fig. 1c), and in quantifying mRNA molecules in mRNA-loaded LNPs. In contrast, fluorescence-based detection may be better suited in elucidating these properties¹⁷.”

Conclusion

“Our work provides impetus for further studies. It demonstrated the feasibility to study complex nanoparticle systems through fluorescence detection modalities, and the methodology described might be well positioned to be adapted to high-performance imaging techniques, such as single-molecule-sensitivity total internal reflection fluorescence (smTIRF) microscopy, for assessment of payload features with structural information. Besides, ...”

4) The conclusion that 80% of particles using the benchmark formulation is empty seems to suggest there isn't enough mRNA and there are excess lipids. The authors varied the concentration of mRNA to lipids but kept the mass ratio of lipid to mRNA the same. Wouldn't simply increasing the amount of mRNA while keeping all lipid composition the same decrease the amounts of empty particles? Would the unencapsulated mRNA increase significantly from the 4%?

Responses: In the manuscript, we varied the nitrogen (on the ionizable lipids)-to-phosphate (on mRNA) ratio (*i.e.*, N/P ratio) from 1 to 12 (**Figure 4** and **Scheme 2**), in which the mRNA concentration was kept the same while the concentration of the ionizable lipid was tuned, and the concentrations of all other lipid components were adjusted proportionally to that of ionizable lipid. We think that decreasing N/P ratio, meaning increasing the relative mass of mRNA to lipids, was to a large extent equivalent to “increasing the mRNA amount” as suggested by the reviewer. We found that the encapsulation efficiency was all nearly 90% under pH 7.4 for N/P ratio from 2 to 12 (**Supplementary Fig. 5**), suggesting that a lipid amount that equaled to an N/P ratio of 2 or higher was sufficient in encapsulating most mRNAs. The presence of a substantial portion of empty LNPs in the benchmark formulation was

Figure 4h of the manuscript for responses to reviewer's comments

indeed fundamentally based on excessive dosage of lipids ($N/P = 6$) in this formulation.

A clear correlation between decreasing N/P ratio (increasing relative mRNA dosage amount) and reduction of the percentage of empty LNPs was observed under pH 4.0 (**Figure 4h**). This finding was intuitively reasonable, as suggested by the reviewer, because the lipids available to form empty LNPs decreased as N/P ratio decreased. It clearly suggested that our CICS method was sensitive enough to capture such changes in the proportion of empty nanoparticles in the LNP system, as indicated by the reviewer.

As the solvent quality drops from 100% EtOH to 25% EtOH when rapid mixing happens, lipids precipitate when the local lipid concentration reaches a critical value (similar to the concept of critical micelle concentration or CMC). This process can happen regardless of presence of mRNA, as evidence from our manuscript (**Figure S1**) and other published reports (**Ref. 5 and 9** in the manuscript) clearly showed that LNPs with a classical structure and size can form without any nucleic acids. When an mRNA molecule is in the proximity of such precipitation, mRNA-ionizable lipid complexation and lipid precipitation happen at the same time, forming mRNA-encapsulated LNPs. Spatial proximity of mRNA before full lipid precipitation was critical to prevent LNPs from being empty, and it is determined by two rate factors below:

- The lipid precipitation rate, or inverse of the characteristic time to form a LNP structure through lipid precipitation, and
- The rate of diffusion of mRNA into lipids, or inverse of the characteristic time for mRNA and lipid to encounter each other.

Both rate factors are function of the (relative) concentrations of mRNA and lipids. When the N/P ratio decreases (increasing mRNA dosage amount), lipid precipitation rate significantly drops. In the prolonged time it needs to form LNPs, mRNAs have a higher rate to encounter precipitating lipids that resulted in reduction of the fraction of empty LNPs (**Figure 4h**); On the other hand, mRNAs have a higher rate to encounter other mRNA molecules during diffusion that resulted in co-complexation and thus increase of mRNA payload in the complexes formed at pH 4.0 (**Figure 4e**). When N/P ratio is as low as 1 or 2, empty LNPs still form at low frequency. We attribute this to unavoidable lipid precipitation without mRNA because the mixing of mRNA and lipids was theoretically impossible to be perfectly uniform. These findings by CICS agree each other on the initial complexation and precipitation step at pH 4.0, and the analyses above resemble features of the kinetics theory of polyelectrolyte complex nanoparticle formation, for which we previously intensely studied (2019, *ACS Nano*, 10.1021/acsnano.9b03334).

Figure 4e of the manuscript for responses to reviewer's comments

However, it was interesting to see that empty LNP fraction did not differ significantly across different N/P ratios at pH 7.4 (Fig. 4h). This may be a result of multiple implications:

- The composition drifts during dialysis from pH 4.0 to pH 7.4 identified by our CICS assessments involve splitting of empty LNPs and subsequent stabilization of the structure by surface PEG lipid. This might have prevented merging of mRNA complexes into empty LNPs during dialysis at a low N/P ratio.
- As shown in **Figure 4j**, when N/P ratio increases (decreasing relative mRNA amount), the number concentration of empty LNPs increased due to obvious reasons of excessive lipids forming LNPs without mRNA. On the other hand, however, the number concentration of mRNA-loaded LNPs also increased due to the kinetic theory we laid out in **Scheme 2** resulting in decreasing mRNA payload. The concentration increases in both populations (empty LNPs and mRNA-loaded LNPs) might coincidentally be proportional to each other, to give an overall consistent number fraction of empty LNPs across different N/P ratios.

Figure 4j of the manuscript for responses to reviewer's comments

In conclusion, we found these analyses based on the data in our manuscript agree with the reasonable, intuitive assumptions proposed by the reviewer on formation of empty LNPs, and altogether support the general sensitivity of capturing the changes in the LNP system by CICS. We thank the reviewer for revisiting this point as we do feel that we did not offer sufficient explanations in the manuscript regarding this point. We therefore have added the following discussions into the section “*Fraction of empty LNPs*” of the revised manuscript:

...“A decrease in N/P ratio reduced the fraction of empty LNPs formed at pH 4.0 (Fig. 4h), highlighting the critical role of relative molar ratio of mRNA to lipids in determining the rate of lipid precipitation alone without mRNA, and the rates of mRNA-ionizable lipid complexation and concurrent lipid co-precipitation. However, the fraction of empty LNPs did not decrease with N/P ratio for these LNP formulations measured at pH 7.4. We attribute this to concurrent effects (Fig. 4j) of a decrease in the concentration of empty LNPs prepared at a lower N/P ratio due to less excessive lipids, and a decrease in the concentration of mRNA-loaded LNPs due to an increased mRNA payload (Scheme 2).”...

Reviewer #2

The authors have provided detailed responses to reviewer's comments, however they have not performed a definitive experiment to verify their results using an independent technique. Thus I am not entirely convinced that the fundamental finding that 80% of the LNP are "empty" is correct. However, I do find it highly likely that a substantial portion are. This manuscript will stimulate further investigations in what is clearly and important parameter affecting the potency of LNP mRNA formulations and I think warrants publication.

Responses: We appreciate these comments from the reviewer. We fully agree that the findings reported in our manuscript will need cross examination from various techniques in the future, and it is certainly our goal to stimulate such efforts from labs and companies around the world with distinct expertise.

We developed CICS as a platform technology that is facile to use, resembling the application of dynamic light scattering as a routine characterization for nanoparticle systems. We firmly believe that the information revealed by CICS will be valuable to a variety of nanomedicine applications.

Reviewer #3

Thanks for the authors' work and responses. I have no further questions.

Responses: We would like to thank the reviewer again for the comments and suggestions that led to improvements of the manuscript.

REVIEWERS' COMMENTS

Reviewer #1 (Remarks to the Author):

The authors' additional clarifications are appreciated. I only have two follow-on suggestions:

1) The underlying message of the paper is that "benchmark formulation of LNPs" are 80% empty of mRNA. As the authors mentioned in the response, this is because of "excessive dosage of lipids" and that when lipid/mRNA lipid ratio is lower, 90% of LNPs would have mRNA.

I think there are two points that may be good to clarify in the abstract. First, what "benchmark formulation" mean. Does it mean current LNPs used in vaccines are made using this formulation and therefore 80% of those LNPs are empty? Or it just means it is a common formulation that appear in publications, but not necessarily what is in vaccines at the moment because that has been further optimized or the formulation is unknown because it is proprietary?

Second, clearly explain that 80% empty is because of the excessive dosage of lipids, and when lipid/mRNA ratio is adjusted, most LNPs are in fact occupied.

I think these points are important to address because the first reaction in reading the paper and abstract is that LNPs are mostly empty and non-functional in vaccines, yet people don't know because they don't have a way of measuring it. But if this impression is not correct, then it is important to clarify it in the abstract.

2) If I understand correctly, there is quite a bit of spread in the spillover, and thus the compensation may not work so well here. However, this is not an issue for the experiments in this paper because very few of the events in the real experiments need compensation. If so, then I would suggest the authors remove the discussions of the compensation and simply provide data showing very few events require compensation and thus compensation is not needed to reach the biological conclusion. Otherwise, it is quite confusing, because there are extensive discussions about correcting spillover when the correction actually might not work well here, but then it doesn't really matter if it doesn't work well because it is not needed. The manuscript might read better without the extensive discussions about spillover if that doesn't matter for the experiments presented.

Responses to Additional Reviewer Comments for Manuscript Payload Distribution and Capacity of mRNA Lipid Nanoparticles

Reviewer #1

The authors' additional clarifications are appreciated. I only have two follow-on suggestions:

1) The underlying message of the paper is that "benchmark formulation of LNPs" are 80% empty of mRNA. As the authors mentioned in the response, this is because of "excessive dosage of lipids" and that when lipid/mRNA lipid ratio is lower, 90% of LNPs would have mRNA.

I think there are two points that may be good to clarify in the abstract. First, what "benchmark formulation" mean. Does it mean current LNPs used in vaccines are made using this formulation and therefore 80% of those LNPs are empty? Or it just means it is a common formulation that appear in publications, but not necessarily what is in vaccines at the moment because that has been further optimized or the formulation is unknown because it is proprietary?

Second, clearly explain that 80% empty is because of the excessive dosage of lipids, and when lipid/mRNA ratio is adjusted, most LNPs are in fact occupied.

I think these points are important to address because the first reaction in reading the paper and abstract is that LNPs are mostly empty and non-functional in vaccines, yet people don't know because they don't have a way of measuring it. But if this impression is not correct, then it is important to clarify it in the abstract.

Responses: We appreciate the cautionary notes from the reviewer. To make these messages more specific, we have as suggested revised the relevant section of the abstract as follows:

“By differentiating unencapsulated mRNAs, empty LNPs and mRNA-loaded LNPs via coincidence analysis of fluorescent tags on different LNP components, and quantitatively resolving single-mRNA fluorescence, we revealed that a commonly referenced benchmark formulation using DLin-MC3 as the ionizable lipid contains mostly 2 mRNAs per loaded LNP with a presence of 40%–80% empty LNPs depending on the assembly conditions.”

To elaborate on the specific points that we highlighted in this revision:

- The specification of the benchmark formulation and the used of DLin-MC3 clearly indicates that the formulation that we characterized is NOT the LNP formulation currently being clinically used in the COVID-19 vaccines. The benchmark formulation commonly serves as a positive control in many publications, such as #3, #6, #7, and #24, which we cited in the manuscript. In the sections of “Characterization of a benchmark mRNA LNP formulation” and “Methods”, we clearly specified all lipid components with their ratios and concentrations, as well as detailed experimental procedures to manufacture this formulation. We clarified that in the abstract to avoid misinterpretation and believe that there should be no confusion upon reading our paper regarding what formulation is being discussed. We also made the following revision in Introduction to accurately reflect this definition (Page 4, Lines 72–73).
“This new technique was first applied to characterize a commonly referenced DLin-MC3-based mRNA LNP formulation in the literature (hereafter termed “the benchmark formulation”).”
- The revised description regarding empty LNPs accurately summarized the major conclusion of our paper. We understand that the reviewer suggested that “excessive dosage of lipids” was the major

reason for presence of empty LNPs. This interpretation is correct for pH 4.0 condition during the LNP preparation (the first step, **Fig. 5h**, red curve), but not so for pH 7.4 condition (the final product condition, **Fig. 5h**, blue curve). We have described in detail in the relevant section “Effects of N/P ratio on payload capacity of mRNA LNPs and composition drift during dialysis” that this was due to dynamic behaviors during pH transition from 4.0 to 7.4. To give a scientifically sound and general description in Abstract, we feel that “depending on assembly conditions” would sufficiently convey that the presence of empty LNPs is indeed generated by the unique assembly processes of mRNA LNPs. Logistically, “assembly condition” covers “excessive lipid dosage” and we feel that this more detailed interpretation can be derived after reading Fig. 5 of this manuscript.

- The reviewer commented that 90% LNPs would contain mRNA (therefore 10% empty) by reducing lipid dosage. This was a specific data point when N/P ratio was lowered to 2. However, the benchmark formulation is specifically referred to that with N/P = 6. We therefore checked the data for N/P = 6 and found the range to be 40% to 80% empty LNPs.
- We also have concerns regarding the use of “excessive” in this context. It is a common practice in the mRNA LNP delivery field that an N/P ratio of 4 to 8 is typically used to ensure delivery efficiency (such as in references #2 to #8). The Pfizer-BioNTech and Moderna vaccines use an N/P ratio of 6 (reference here¹), which is the same as the benchmark formulation we characterized. This level of lipid dosage is well accepted by the LNP delivery research field. We therefore do not consider the formulation we characterized had “excessive” lipid dosage.

2) If I understand correctly, there is quite a bit of spread in the spillover, and thus the compensation may not work so well here. However, this is not an issue for the experiments in this paper because very few of the events in the real experiments need compensation. If so, then I would suggest the authors remove the discussions of the compensation and simply provide data showing very few events require compensation and thus compensation is not needed to reach the biological conclusion. Otherwise, it is quite confusing, because there are extensive discussions about correcting spillover when the correction actually might not work well here, but then it doesn't really matter if it doesn't work well because it is not needed. The manuscript might read better without the extensive discussions about spillover if that doesn't matter for the experiments presented.

Responses: The insights from the reviewer are correct, and we agree with the reviewer’s suggestion. We have therefore made the following revisions:

- The original figures showcasing an example of compensation (Fig. 1d and 1e in the last version of the revised manuscript) have been moved to Supplementary Materials, now merged into Fig. S2 together with other figures related to compensation. Instead, we added the cryo-TEM images of the representative LNP preparations with and without labeled mRNA or mRNA (Fig. 1g–i).
- The discussion about compensation in the main text of the manuscript has been simplified:

“The raw data were processed by a thresholding algorithm to identify and quantify these fluorescent bursts. Different species of interest in an LNP formulation were determined by coincidence analysis of the fluorescence bursts (Fig. 1c, Table 1). Fluorescent spillovers across different channels were only occasionally observed in CICS (Supplementary Fig. 2). Compensations with single stain controls were carried out and proved to be effective for this CICS platform (Supplementary Discussion 1), even though we found that the quantification results to be reported throughout Fig. 2 to Fig. 7 were largely insensitive to compensation (Supplementary Table 5).”

- We do not want to completely delete all descriptions related to compensation because this is a routine procedure used in multi-color spectroscopy. We therefore have put detailed discussions, including many data tables that we used to address reviewer's comments, into Supplementary Materials. They are now in Supplementary Discussion 1 and Supplementary Tables 2 to 5. We hope a broad readership who may have concerns over cross-channel fluorescent signal spillover can find this information helpful, and therefore understand the levels of accuracy and reliability of our data and conclusions.

References for Response Letter

1. Schoenmaker, L. et al. mRNA-lipid nanoparticle COVID-19 vaccines: Structure and stability. *International Journal of Pharmaceutics* **601**, 120586 (2021).